# Spatial subsidies drive sweet spots of tropical marine biomass production

Renato A. Morais[1,2]*, Alexandre C. Siqueira[1,2], Patrick F. Smallhorn-West[2,3], David R. Bellwood[1,2]

1 Research Hub for Coral Reef Ecosystem Functions, College of Science and Engineering, James Cook University, Townsville, Australia, 2 ARC Centre of Excellence for Coral Reef Studies, James Cook University, Townsville, Australia, 3 WorldFish, Bayan Lepas, Malaysia

* renato.morais@my.jcu.edu.au

**Data Availability Statement:** All data and code required to reproduce the figures and tables of this manuscript are available from its Zenodo repository, accessible at: https://doi.org/10.5281/zenodo.5540102.

## Abstract

Spatial subsidies increase local productivity and boost consumer abundance beyond the limits imposed by local resources. In marine ecosystems, deeper water and open ocean subsidies promote animal aggregations and enhance biomass that is critical for human harvesting. However, the scale of this phenomenon in tropical marine systems remains unknown. Here, we integrate a detailed assessment of biomass production in 3 key locations, spanning a major biodiversity and abundance gradient, with an ocean-scale dataset of fish counts to predict the extent and magnitude of plankton subsidies to fishes on coral reefs. We show that planktivorous fish-mediated spatial subsidies are widespread across the Indian and Pacific oceans and drive local spikes in biomass production that can lead to extreme productivity, up to 30 kg ha$^{-1}$ day$^{-1}$. Plankton subsidies form the basis of productivity "sweet spots" where planktivores provide more than 50% of the total fish production, more than all other trophic groups combined. These sweet spots operate at regional, site, and smaller local scales. By harvesting oceanic productivity, planktivores bypass spatial constraints imposed by local primary productivity, creating "oases" of tropical fish biomass that are accessible to humans.

## Introduction

Ecosystems are frequently connected by flows of energy and material [1–3]. These flows occur via the movement of resources (i.e., nutrients, detritus, or prey) or consumers across ecosystem boundaries, characterising spatial subsidies [2,3]. Spatial subsidies involve at least 1 donor and 1 recipient ecosystem. From the perspective of the recipient ecosystem, spatial subsidies (external or "allochthonous inputs") increase local productivity and boost consumer abundance beyond the limits imposed by local resources [2,4–6]. Such subsidies have been found to be key elements of food webs in streams [7], lakes [8], islands [9], and on tropical coral reefs [10–14] but are likely to operate wherever discrete biological communities interact [1,2].

Coral reefs support consumer communities that span multiple trophic levels, attain high biomass, and provide critical food resources for humans. External inputs of nutrients, detritus,

**Funding:** Funded by the Australian Research Council through a Laureate Fellowship (FL190100062 to DRB). Also contributed to funding: James Cook University (Postgraduate Research Scholarship to RAM, ACS and PSW, and HDR Competitive Research Training Grant to RAM), the Lizard Island Reef Research Foundation (Lizard Island Doctoral Fellowship to RAM), the Ocean Geographic Society (Elysium Heart of the Coral Triangle Expedition to RAM), the National Geographic Society (CP-137ER-17 to PSW). The funders had no role in study design, data collection and analysis, decision to publish, or preparation of the manuscript.

**Competing interests:** The authors have declared that no competing interests exist.

**Abbreviations:** ELPD, expected log predictive density; GC, generalised macrocarnivores; GLM, generalised linear model; HD, herbivores; HPD, high posterior density; IAA, Indo-Australian Archipelago; IN, invertivores; nMDS, nonmetric multidimensional scaling; NPP, net primary productivity; NUTS, No-U-Turn Sampler; OM, omnivores; PB, productivity–biomass; PCoA, principal coordinate analysis; PK, zooplanktivores; RLS, Reef Life Survey; VBGM, Von Bertalanffy Growth Model.

and plankton from a myriad of oceanographic mechanisms [14], and interhabitat consumer mobility, connect reefs to adjacent ecosystems across seascapes (e.g., mangroves, seagrass beds, open ocean) [15–19]. These processes provide energy and nutrient subsidies and channel the productivity from large unconstrained areas to the strongly delineated coral reef structure. As a result, subsidies allow nonconventional, "top-heavy," or "convex" trophic structures to emerge [5,6,19,20]. Yet, external spatial subsidies have only recently begun to be more widely contemplated as a mechanism to explain patterns in the distribution, resource use, and the trophic structure of coral reef consumers [19–22]. These trophic assessments have revealed a potential direct link between spatial subsidies (i.e., plankton) and increased fish biomass [23] and productivity [11], suggesting a prominent role for energetic and nutrient connectivity in boosting or driving highly productive consumer communities on coral reefs.

Spatial subsidies may also help to explain human behaviour in tropical reef fisheries. Fishers do not fish at random [24–27] (**Fig 1A**). Given their limitations (e.g., technology, economic, individual ability), fishers will actively search for the best fishing grounds, "sweet spots" of fish concentration where effort or risks are minimised and rewards are maximised [25,28]. We use the vernacular term "sweet spots" to describe these areas of fish concentration where rewards are likely to be optimal (for a different use in the context of fisheries, see [29]). While "optimal rewards" are context dependent [25,27], locations that concentrate target fish biomass production (**Fig 1B**) will likely also support increased yields (or profit) relative to the rest of the seascape [26,28,30,31]. Plankton inputs, for example, tend to drive planktivorous and predatory fish aggregations in suitable habitats [11,22,23,32,33], with potential flow-on effects for important coastal and reef fisheries [34–38]. But how common are these "sweet spots" of fish concentration, both globally and locally, and what is the role of spatial subsidies in shaping their occurrence?

By dissecting the trophic structure of reef fish assemblages, we reveal the extent and magnitude of spatial subsidies to coral reef fish consumers. We focus solely on plankton subsidies incorporated by planktivorous fishes. We do not consider indirect trophic links that may also result in these subsidies being incorporated by reef communities (e.g., predators of planktivorous fishes; see Methods). Specifically, we combine a global database of reef fish abundances (1,035 sites in 32 ecoregions) with focal surveys providing high-resolution estimates of fish productivity (325 survey areas in 3 locations) to show that (1) planktivorous fishes, as key vectors of subsidies from offshore waters to coral reefs [2,39,40], can underpin biomass production rates that outpace all other fish guilds combined; and that (2) spatial subsidies mediated by planktivorous fishes are geographically widespread and drive "sweet spots" of high biomass production on coral reefs. These sweet spots of fish biomass production (indicated by a majority [50% or more] of fish productivity comprised by planktivores) are a common feature of Indo-Pacific coral reefs. Yet, these sweet spots are locally concentrated within the seascape, occurring only in specific sites within regions. Importantly, sweet spots with extreme fish productivities (15 to 30 kg ha$^{-1}$ day$^{-1}$) were exclusively associated with dominance by planktivorous fishes, and no other guild.

## Results

Planktivorous reef fishes are a ubiquitous and dominant element of Indo-Pacific coral reefs. This is evidenced by planktivores accounting for the largest proportion of fish abundance in most of the 32 ecoregions investigated in the ocean-scale dataset (see Methods; **Fig 2A**). Across all sites, fish abundances exceeding 1,000 ind 250 m$^{-2}$ were typically comprised of 75% or more of planktivores. While the relative abundance of herbivores, omnivores, invertivores, and generalised macrocarnivores displayed either a negative or a hump-shaped association

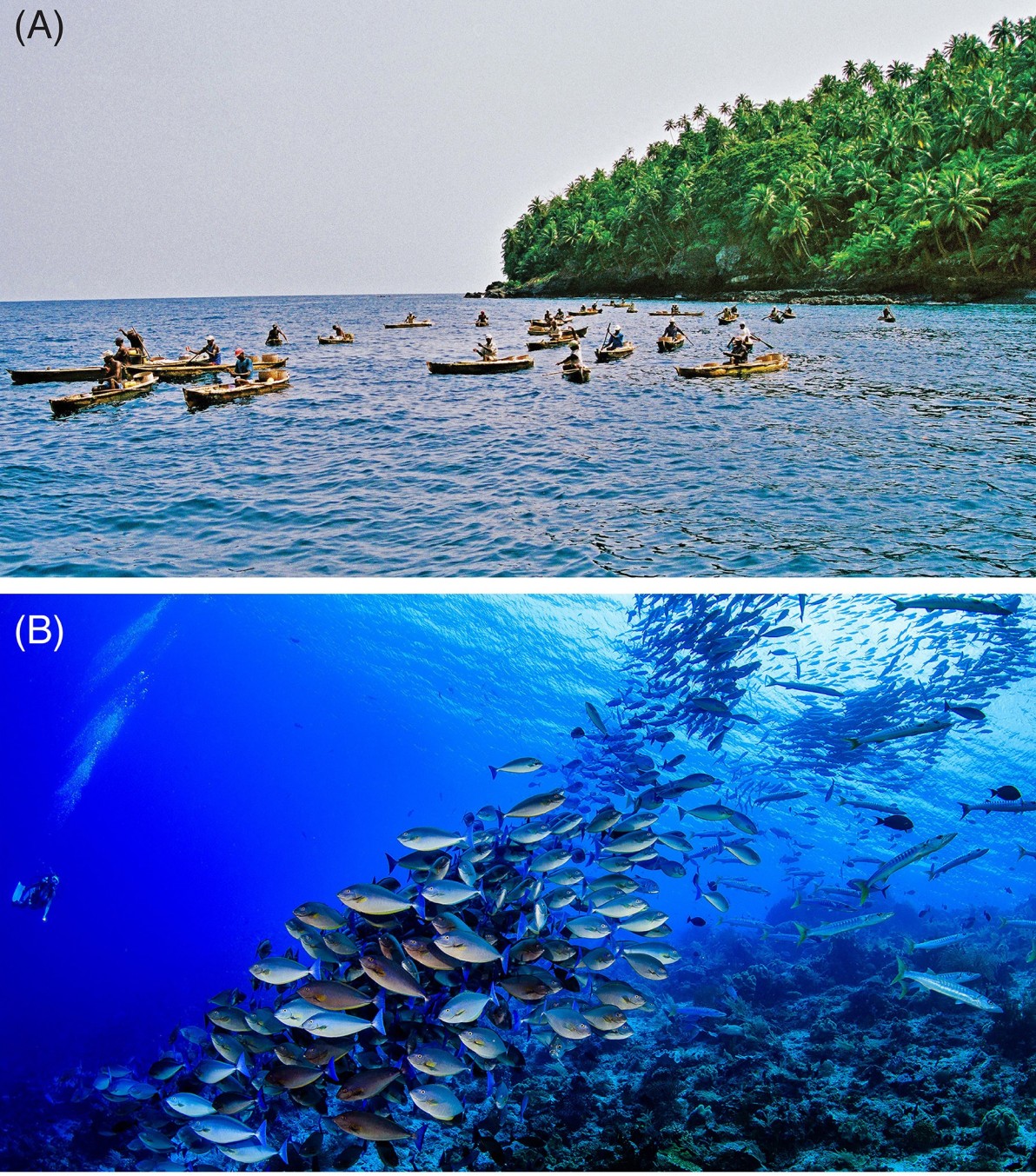

**Fig 1. Neither fishers nor fishes are randomly distributed across reef scapes. (A)** By searching and concentrating on the best, "sweetest" fishing spots, fishers intend to maximise their catch relative to effort or risk. **(B)** Here, we show that plankton subsidies can, and do, underpin the occurrence of "sweet spots" of fish productivity on coral reefs at a global scale, sustaining productive assemblage not only of planktivorous reef fishes, but also of their predators. *Photo credits: (A) João Luiz Gasparini, National Geographic Society; (B) Yen Yi Lee, Coral Reef Image Bank.*

with total fish abundance, that of planktivores increased almost monotonically (**Fig 2B**). This involved a sharp rise in planktivore proportions at intermediate to high fish abundances ($>300$ and $<3,000$ ind 250 m$^{-2}$), although with substantial variability (**Fig 2B**). As a result,

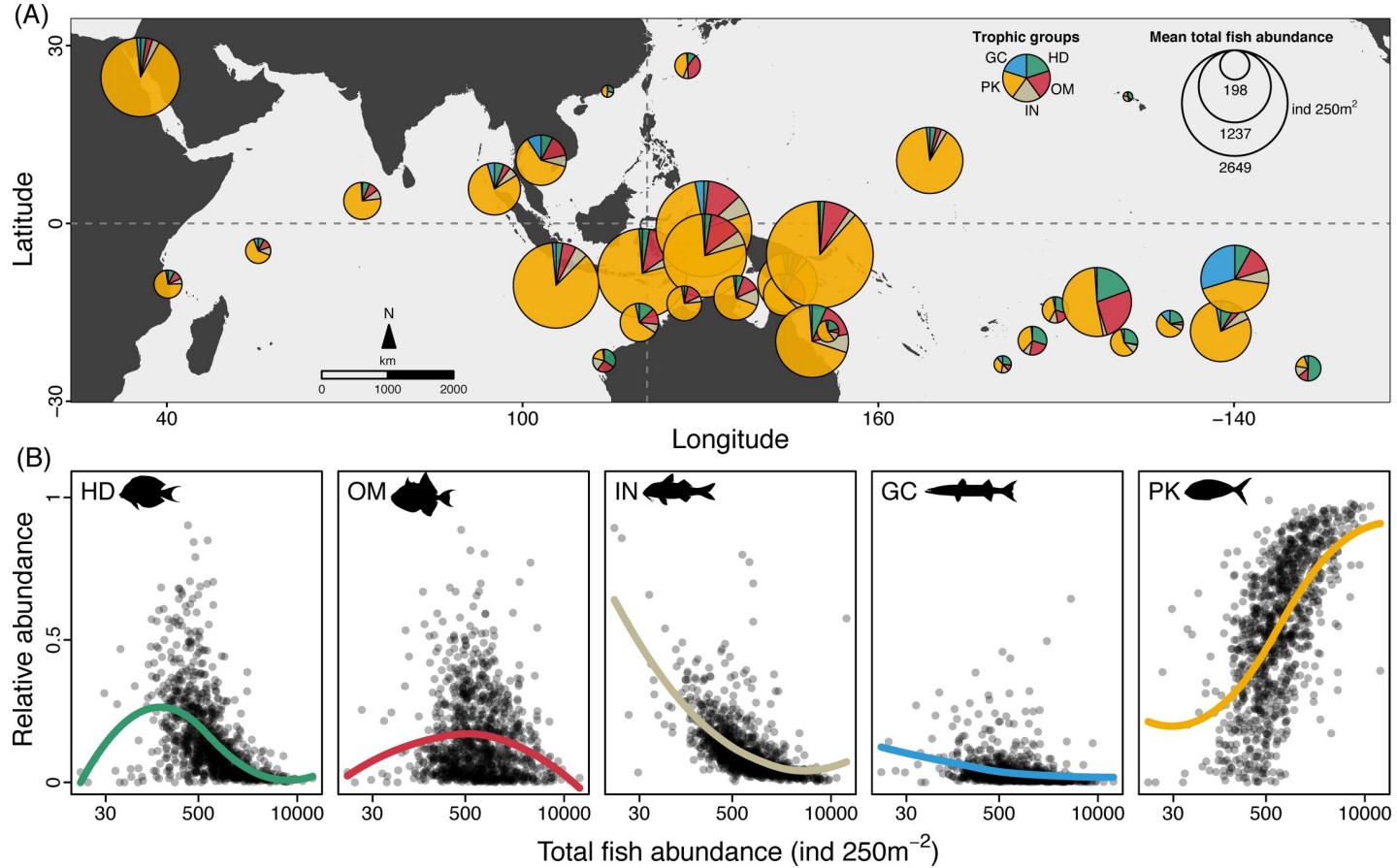

**Fig 2. The trophic structure of Indo-Pacific reef fish assemblages. (A)** Reef fish abundances across 32 ecoregions, from the Western Indian to the Central Pacific, were clearly dominated by planktivores (in yellow). Pies indicate proportional mean abundance for ecoregions from the RLS dataset [41]. **(B)** From the 5 reef fish trophic groups evaluated, PK are the only one where proportional abundance increases as total fish abundance increases. Dots represent mean abundances for each of the 1,035 sites, and trend lines are polynomial regression fits (LOESS smoothers). Note the log-scale in the x-axis of panels in (B). Map source: Natural Earth via the maps and mapdata packages in R. Numerical values underlying this figure are provided in "Morais_et_al_Fig 02_DataS1.R," available from https://doi.org/10.5281/zenodo.5540102. GC, generalised macrocarnivores; HD, herbivores/detritivores; IN, invertivores; OM, omnivores; PK, planktivores; RLS, Reef Life Survey.

planktivores overwhelmingly dominated sites with high fish abundance ($>$3,000 ind 250 m$^{-2}$), often comprising 80% or more of all individuals. But what are the energetic implications of these high abundances of planktivorous fishes for tropical coral reefs?

To answer this question, we examined in detail 3 selected locations spanning a major biodiversity and abundance gradient [42]. The 3 locations, Raja Ampat (Indonesia, in the core Indo-Australian Archipelago [IAA]), Lizard Island (Great Barrier Reef, in the periphery of the IAA), and Ha'apai (Tonga, in the Central Pacific), replicated the extensive differences in total fish abundance and planktivore proportion from the ocean-scale dataset (**S1 Fig**, **S1 Text**). Total fish productivity in Raja Ampat (11.94 kg ha$^{-1}$ day$^{-1}$ [wet weight]) was over 3 times higher than at Lizard Island (3.58 kg ha$^{-1}$ day$^{-1}$; $\beta_{pro}$ = 3.34, 95% high posterior density [HPD] interval = 2.24 to 4.66) and more than 6 times higher than in Ha'apai (1.89 kg ha$^{-1}$ day$^{-1}$; $\beta_{pro}$ = 6.33, HPD = 4.56 to 8.51). Differences in standing biomass between locations were less extreme, being 55% higher in Raja Ampat (4.15 t ha$^{-1}$) compared with Lizard Island (2.68 t ha$^{-1}$, $\beta_{bio}$ = 1.55, HPD = 0.94 to 2.34) and 199% higher in Raja Ampat than in Ha'apai (1.39 t ha$^{-1}$, $\beta_{bio}$ = 2.99, HPD = 1.99 to 4.29). The 3 locations had a similar rate of increase in productivity per unit biomass (i.e., slope; **Fig 3A**), yet there were substantial differences in their basal

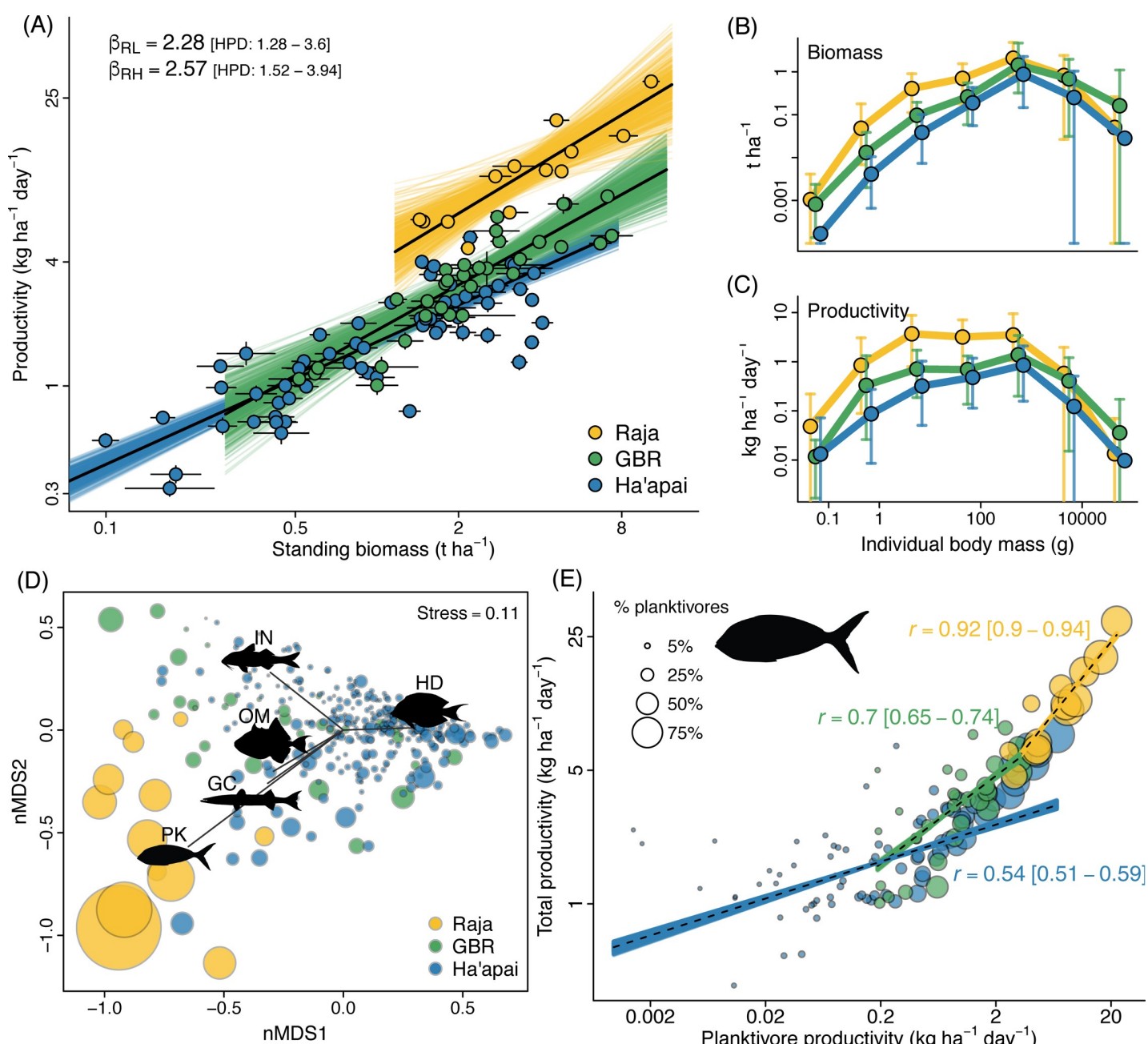

**Fig 3. Distinct reef fish productivity–biomass relationships among the 3 locations examined arise from similar size structures but different trophic structures.** (A) Regression lines for all locations had a similar slope but different intercepts. Dots are the bootstrap medians for survey areas, and segments delimit the 95% bootstrap quantile intervals (see Methods). Coloured lines represent 500 draws from the Bayesian posterior distributions and black lines represent the posterior medians. (B) Biomass and (C) productivity size distributions, with median (dots) and 90% bootstrap quantile intervals. (D) An nMDS biplot showing trophic groups as vectors and survey areas as circles scaled proportionally to total fish productivity. (E) Relationship between the productivity of PK and total fish productivity in the 3 locations. Trendlines depict 300 bootstrapped Pearson's correlation r values (coloured lines) and median r (dashed lines) back calculated from standardised variables to productivity units (see Methods and S3 Fig) for each location. Circles are survey areas and are scaled proportionally to relative PK productivity. To improve clarity, only 60 randomly selected survey areas out of 284 are depicted for Ha'apai in (A) and (E). The same plots with all data points unscaled are available as S2 Fig. Note the log-scale in the axes of panels (A), (B), (C), and (E). Numerical values underlying this figure are provided in "Morais_et_al_Fig 03_FigS2.R," available from https://doi.org/10.5281/zenodo.5540102. GBR, Great Barrier Reef; GC, generalised carnivores; HD, herbivores/detritivores; IN, invertivores; nMDS, nonmetric multidimensional scaling; OM, omnivores; PK, planktivores.

level of productivity (i.e., intercept; **Fig 3A**, **S1 Table**). For the same values of biomass, fish productivities were 2.3 times larger in Raja Ampat compared to Lizard Island ($\beta_{pbr}$ = 2.28, HPD = 1.28 to 3.60; **Fig 3A**, **S2 Fig**) and almost 2.6 times larger than in Ha'apai ($\beta_{pbr}$ = 2.57, HPD = 1.52 to 3.94) (for coefficient values, see **S1 Table**).

These marked disparities in productivity between Raja Ampat, Lizard Island, and Ha'apai appeared not to be merely due to biodiversity effects. Indeed, differences in α, β, and γ species diversities between the 3 locations were insufficient to explain distinctions in productivity (**S1 Text**). These disparities in productivity also appeared not to be determined by differential rates of human exploitation in the locations surveyed (e.g., [43]), as evidenced by similarities in fish size structure among locations. There was large variability and widespread overlap in quantile intervals between locations for both size-specific biomass and productivity, but the most productive region (Raja Ampat) had similar or higher productivity compared with the other locations in all size classes (**Fig 3B and 3C**). Additional support was provided by a lack of relationship between a proxy of human exploitation rates and either biomass or productivity (**S1 Text**, *Testing for potential effects of human exploitation*).

In contrast to the lack of explanatory power in diversity or size structures, there was a strong distinction among regions in the trophic structure of reef fish assemblages (**Fig 3D**). The trophic structure at Lizard Island and Ha'apai was mainly associated with herbivores and invertivores, but in Raja Ampat, it was dominated by generalised carnivores, omnivores, and, especially, by planktivores. The relationship between planktivore and total fish productivity was stronger, steeper, and had higher values for both variables in Raja Ampat compared to Lizard Island and Ha'apai (**Fig 3E**). Extreme fish productivities (15 to 30 kg ha$^{-1}$ day$^{-1}$) were exclusively driven by extreme planktivore productivities (12 to 22 kg ha$^{-1}$ day$^{-1}$). Nevertheless, planktivore and total fish productivity were highly correlated, and the highest productivity values were frequently associated with the highest planktivore productivities for all locations, not just Raja Ampat (**Fig 3E**, **S3 Fig**; Raja Ampat $r$ = 0.92 [0.90 to 0.94]; Lizard Island $r$ = 0.70 [0.65 to 0.74]; Ha'apai $r$ = 0.54 [0.51 to 0.59]; median Pearson's $r$ [lower and upper 95% bootstrap quantiles]). Furthermore, as the relative productivity of planktivores increased, total fish productivity increased almost exponentially (**S4 Fig**). No other trophic groups exhibited the same simultaneously tight, steep, and sequential relationship with total fish productivity as planktivores (**S3 Fig**).

To assess the geographic extent of planktivore-mediated subsidies, we explored the energetic implications of high planktivore productivity (revealed by the analysis of the 3 focal locations) within the ocean-scale dataset. First, we searched for potential predictors of the proportional productivity of planktivores in these 3 locations (see Methods). We found that this was predicted with high precision by planktivore species abundance and maximum size, current velocity, and pelagic net primary productivity (NPP) (Bayesian $R^2$ = 0.80; HPD = 0.77 to 0.82). Relative planktivore productivity increased with planktivore abundance. This increase was steeper with larger species present, slower currents, and higher NPP (**S5 Fig**). For average values of these variables, planktivore abundances of 500 to 1,000 ind 100 m$^{-2}$ meant more than 50% of total productivity would be accounted for by planktivores, more than all other trophic groups combined (**S5 Fig**). Notably, these "sweet spots" (i.e., sites with planktivores contributing >50% of the total productivity, which was generally high), occurred in all 3 locations, but they comprised only 39 out of a total of 325 survey areas.

We then used this model to predict the expected contributions of planktivores to productivity in the ocean-scale dataset. We found evidence for widespread planktivore subsidies, with a mean expected planktivore contribution to total productivity of 28% (HPD = 26% to 30%) across 1,028 sites (see Methods; **Fig 4**). However, this concealed important spatial variation (**Fig 4**). Despite small differences between oceans (Western Indian = 33%; Central Indo-

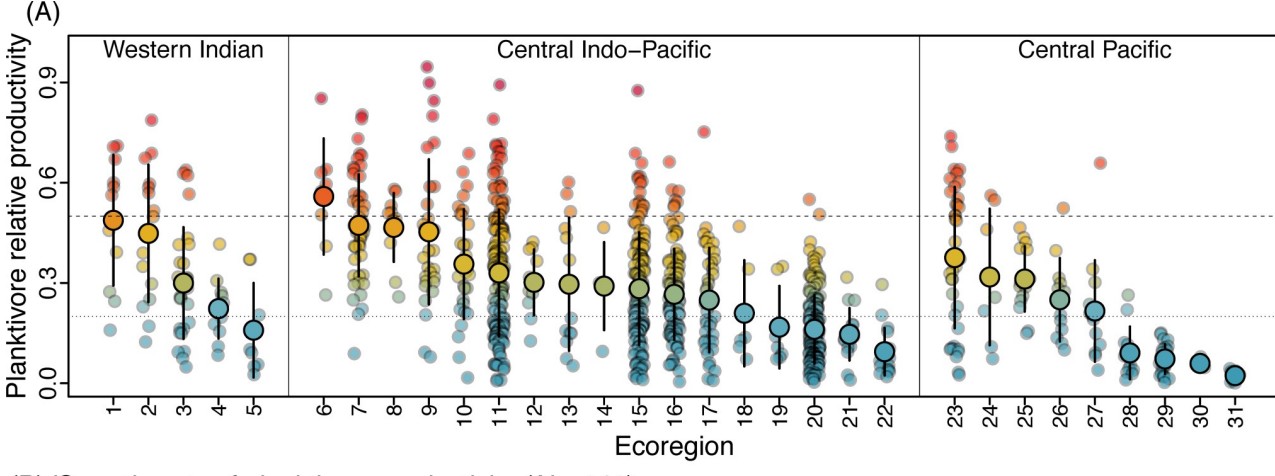

(A)

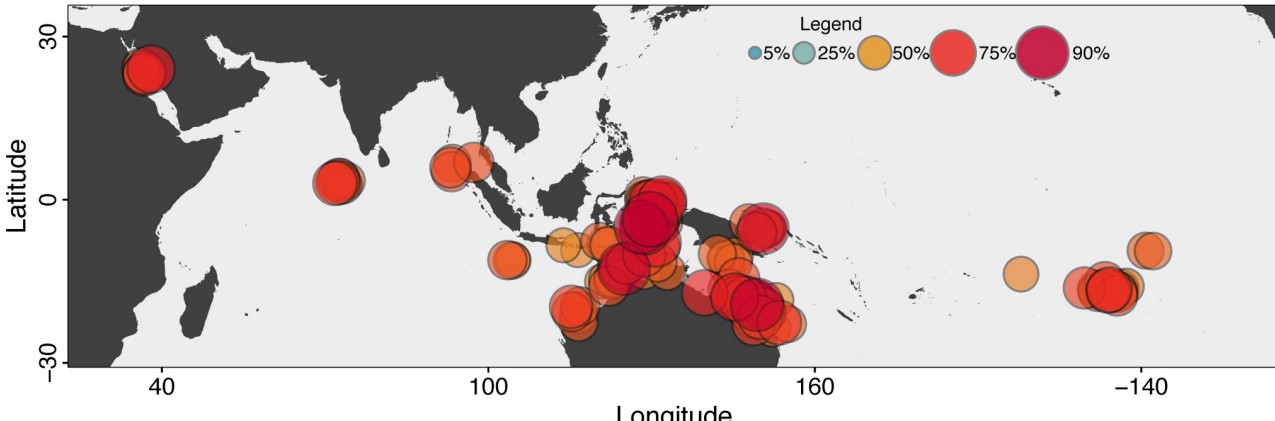

(B) 'Sweet' spots of planktivore productivity (*N* = 141)

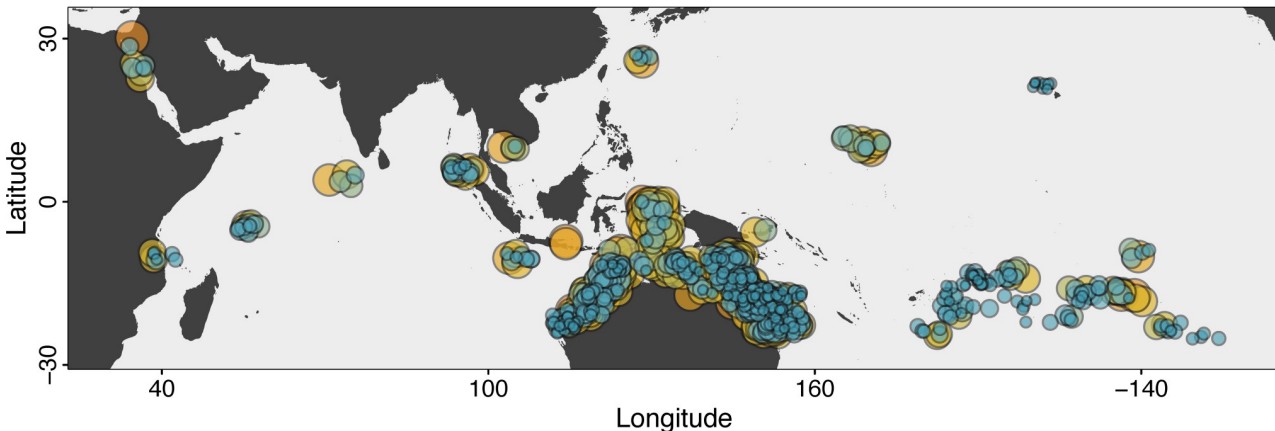

(C) Low planktivore productivity sites (*N* = 887)

**Fig 4. The geography of planktivore contributions to fish productivity on Indo-Pacific coral reefs. (A)** The distribution of predicted proportional planktivore productivity for 1,028 sites in 31 ecoregions distributed across the Western Indian, Central Indo-Pacific, and Central Pacific Oceans. Larger circles are means, and intervals represent standard deviations. Dashed line marks the "sweet spot" threshold (50% or more of total productivity), and the dotted line marks the expected proportion comprised by any single trophic group (20% of total productivity). Region labels are in the **S1 Data**. **(B)** The geographic location of "sweet spots" (>50% of total) and **(C)** low (<50% of total) planktivore proportional productivity. In all plots, colours range from low (blue shades) to high (red shades) predicted relative planktivorous fish productivity. Map source: Natural Earth via the maps and mapdata packages in R. Numerical values underlying this figure are provided in "Morais_et_al_Fig 04.R," available from https://doi.org/10.5281/zenodo.5540102.

Pacific = 28%; Central Pacific = 23%), 18 out of 31 ecoregions in the 3 oceans had sites where >50% of total productivity was driven by planktivores. These sweet spots of high planktivore productivity characterised areas of high total productivity and strong plankton subsidies (**Fig 3**), with a robust statistical relationship between the 2 attributes (**S4** and **S5 Figs**). Sweet spots were restricted to just 141 of the 1,028 sites, although almost 60% of all the sites (603 of 1,028 sites located in all but 4 ecoregions) had at least 20% planktivore-mediated productivity (thin dotted line, **Fig 4A**). Overall, planktivore-mediated subsidies were ubiquitous and drove the widespread occurrence of sweet spots at regional (18 of 31 ecoregions) and local (141 of 1,028 sites in the ocean-scale dataset) scales, and even in more localised areas (39 of 325 survey areas in the regional dataset). In exceptional cases (12/141 sweet spots in the ocean dataset and 7/39 sweet spots in the regional dataset), planktivores comprised between 75% and 95% of total fish productivity.

## Discussion

We documented the overwhelming role of planktivorous fishes in driving "sweet spots" of biomass production on tropical coral reefs. This recognition hinges on a series of observations (1) that high abundances of planktivorous fishes can be found on reefs in almost any region, from the Red Sea to the eastern South Pacific; (2) that planktivores dominate reef fish assemblages characterised by high abundance throughout the Indo-Pacific Ocean, with these high abundances being driven by the planktivores themselves; (3) that planktivore contributions to total fish productivity of 50% or more are a relatively common, and widespread, phenomenon on Indo-Pacific coral reefs; (4) that these high-abundance planktivore assemblages can sustain exceptionally high biomass production rates (up to 22 kg ha$^{-1}$ day$^{-1}$), driving total fish productivities that may exceed 30 kg ha$^{-1}$ day$^{-1}$; and (5) that planktivore productivity was highly correlated with total fish productivity, not only between locations, but also within the 3 selected locations.

These high planktivore productivities were not associated with low predator abundance, as would be expected under trophic release from the overfishing of predators (**S1 Text**, *Testing for potential effects of human exploitation*). Indeed, there was a positive relationship between the productivity of planktivores and that of predatory fishes, suggesting bottom-up, rather than top-down effects. No other trophic group on coral reefs attained such extreme productivities as planktivores or were so intimately associated with overall fish productivity, including their predators. These results indicate that planktivores are fundamental contributors to the critical ecosystem function of fish biomass production on tropical reefs across the globe.

Importantly, by vectoring external productivity in the form of plankton, planktivores provide a key seascape function that allows coral reefs to escape spatial constraints dictated by internal photosynthesis. Indeed, planktivory has been a common destination in coral reef fish evolutionary transitions [44], with planktivores contributing disproportionally to the global marine biodiversity gradient [42]. Because of thermodynamic principles, internal primary productivity on ecosystems is bound to constrain the production of consumer biomass, unless it is circumvented by mobile consumers feeding on adjacent habitats (e.g., [2,17,19,21]), or inputs of food particles [11,23,40,45]. As our data revealed, planktivores stand as the driving force behind exceptionally high fish productivities (up to 22 out of 30 kg ha$^{-1}$ day$^{-1}$ in Raja Ampat). The productivity of all the other trophic groups combined amounted to, on average, just 1.4 to 3.6 kg ha$^{-1}$ day$^{-1}$ in the 3 locations investigated. These values are consistent with a conservative estimate of consumer productivity relying exclusively on internal benthic photosynthesis: between 0.7 and 5.8 kg ha$^{-1}$ day$^{-1}$ (based on the usual range of benthic primary productivity on coral reefs, see **S1 Text**). Even considering the highest benthic productivities reported, and

largely disregarding trophic inefficiencies due to thermodynamic constraints, a ceiling of 11.7 to 13.3 kg ha$^{-1}$ day$^{-1}$ of endogenous fish production is the maximum that should be expected on coral reefs (**S1 Text**). These values still fall far short of the maximum observed planktivore productivity (22.4 kg ha$^{-1}$ day$^{-1}$). Breaking free from the spatial constraints imposed by reliance on internal production, reef fronts may intercept flows of large volumes of nutrient- or plankton-rich waters [16,39]. Each m$^2$ of forereef occupied by planktivorous fishes in Raja Ampat, for example, is estimated to require the phytoplankton productivity of 1,370 to 7,143 m$^2$ of surface pelagic waters, every day (**S1 Text**). These planktivore assemblages illustrate the fact that, in harvesting productivity carried by ocean currents from larger areas, planktivorous fishes considerably expand the energetic and nutrient footprint of coral reef consumers.

We quantify external subsidies indirectly by estimating the net somatic productivity of planktivorous reef fishes from underwater visual counts. Underwater counts rely on visual fish detection, which has been highlighted as a potential source of methodological bias (e.g., [46–49]). Some planktivorous fishes, for example, have been found to have higher detectability than other reef fish groups [46], likely a consequence of their conspicuous nature and relatively high site association. Large schools of planktivores were a prominent feature of the hyper productive fish assemblages we highlight here, yet these involved both small, site-associated (e.g., damselfishes) and large and/or highly mobile groups (e.g., fusiliers). These highly mobile planktivores often feed and spend significant time away from the reef (e.g., [39,50]), likely reducing, not increasing, their detectability, particularly where they are most abundant. Furthermore, field evidence indicates that larger school sizes per se do not increase fish detectability in visual counts [51]. Altogether, this suggests that the abundances recorded herein are conservative and that actual planktivore abundances may be considerably higher.

On coral reefs, planktivorous fishes are overwhelmingly sustained by pelagic plankton (including oceanic, near-reef, and deeper water plankton [14]) (e.g., [39,52,53]). To eliminate any reef inputs, "planktivorous" species that are also reliant on reef-based material were specifically excluded (see Methods) and do not contribute towards our conclusions. Furthermore, we do not consider indirect trophic interactions that often result in pelagic energy incorporated by other guilds of fishes (e.g., [40,54–56]). Thus, our estimates of net production of planktivorous fishes provide a robust, time-integrated, yet conservative measure of realised spatial subsidies (i.e., after plankton consumption) to coral reef fish assemblages.

Pelagic plankton productivity underpins biological oases in barren tropical oceans and determines global fisheries catches [14,57]. In tropical reef systems, pelagic plankton promotes fish aggregations and enhances fish biomass [20,22,23], but their supply depends on both large-scale oceanographic processes and localised water flows [14,39,58] (see **S1 Text**). The "sweet spots" we identify here feature conditions that coincide to maximise local planktivore productivity, to a point where they outpace the production of all other fish groups combined. These sweet spots operate across a variety of spatial scales: in most regions, in sites with favourable conditions within regions (**Fig 4**), and even in smaller survey areas within localities with generally unfavourable conditions (**S5A Fig**). Reef fishers actively search for locations that balance reward, effort, and risk [25,28]. These are likely to coincide with the plankton subsidy "sweet spots" we identify as dominating local reef fish biomass production, as reported previously for specific locations (e.g., [23,34,59,60]). Critically, the planktivorous fishes underscoring this seascape function include both small, often colourful fishes (important prey of targeted reef predators), and medium to large-sized, school-forming reef fishes that may be major components of tropical reef fisheries yields (e.g., fusiliers [Caesioninae], planktivorous surgeonfishes and unicornfishes [Acanthuridae], and baitfish [Clupeidae] [34,35,60–63]). Furthermore, we found a positive relationship between the productivity of planktivores and of predatory fishes (**S1 Text**, **S6 Fig**), which constitutes evidence supporting the hypothesis that

planktivores may compose a major source of prey for predatory fishes. Thus, plankton pathways produce reef fish biomass that is both directly harvestable and supports higher trophic-level fisheries resources.

Some tropical fisheries are unlikely to benefit, at least directly, from the plankton subsidies we recognise here. Trap and seine net fisheries, for example, are frequently dominated by herbivorous fishes (e.g., [64–66]), which tend to respond to habitat and/or benthic processes [65,67,68]. However, given their intrinsically strong relationship with total fish productivity, planktivore sweet spots are likely to be disproportionally important for many reef-based fisheries [35–38]. Nevertheless, it is conceivable that seascapes that feature favourable conditions for both plankton subsidy and herbivore sweet spots will offer maximised yield opportunities for tropical reef fisheries.

In summary, we provide evidence that spatial subsidies mediated by planktivorous fishes are ubiquitous, key elements of the energetic functioning of coral reefs. By harvesting productivity from larger areas, planktivores bypass spatial constraints imposed by internal photosynthesis and can drive extreme biomass production, unmatched by any other trophic groups. In doing so, they underpin "sweet spots" of potential fisheries productivity [43]. These spots, in effect, concentrate dispersed pelagic plankton productivity into condensed fish biomass "oases" on tropical reefs, where they are accessible for people. As coral reef structures degrade [69] and global pelagic photosynthesis is predicted to decline [70,71], sweet spots that focus dwindling pelagic productivity are likely to become even more important for the future of tropical reef fisheries.

## Methods

### Ethics statement

All research activities were conducted in accordance with James Cook University Animal Ethic Guidelines (permit approval numbers: A2775 to RAM and A2454 to PSW).

### Study design and survey datasets

To assess the trophic structure of reef fish assemblages and the role of planktivore-mediated subsidies, we explored 2 datasets of underwater visual surveys. These consisted of an ocean basin scale dataset spanning most of the Indo-Pacific Ocean and a regional scale dataset focused on 3 key locations. For the ocean basin scale, we used the open access database of reef fish counts from the Reef Life Survey (RLS; [72]), which includes data on abundance and species identity. The complete description of the visual survey method used is available in Edgar and Stuart-Smith [41], but, in brief, it consists in a pair of divers simultaneously conducting fish counts on parallel "blocks." Each belt transect is the sum of the 2 blocks, each comprising an area of 50 m × 5 m, in a total area of 500 m$^2$ per transect. The complete RLS dataset was filtered online to include only tropical locations (between the latitudes of 30° S and 30° N) in the Indo-Pacific (approximately between the longitudes of 35° E and 130° W). We then excluded nonquantitative records (i.e., variable "Block" = "0"), as well as surveys shallower than 4 m or deeper than 15 m depth. Since reef zone information is not available from the RLS dataset, this step was required to minimise the chance of surveys located in back reef zones, downstream from currents and naturally scarce in plankton and planktivorous fishes (e.g., [11,73]). From the resulting set of surveys, we only retained ecoregions (sensu Spalding and colleagues [74]) that encompassed a minimum of 4 transects. Counts were then averaged, rather than summed, between the 2 blocks to obtain a single count relative to an area of 250 m$^2$ per transect. We also averaged across transects at each site for each ecoregion. The final dataset comprised 32 ecoregions and 1,035 unique sites (**Fig 2A**, **S1 Data**).

At the region scale, we surveyed 3 key locations in the Indo-Pacific: Raja Ampat and the Banda Sea (hereafter "Raja Ampat"), in eastern Indonesia; Lizard Island, in the northern Great Barrier Reef, Australia; and Ha'apai, in the central part of Tonga (S1 Fig). Surveys in Raja Ampat and Lizard Island were conducted following a 4-phase design, with each phase targeting specific sizes, mobility categories, and common behavioural responses of fishes in progressively smaller areas. The complete method is described in [11] and involves identifying, counting, and estimating the body size (total length, in cm) of (1) large and mobile, conspicuous fishes within an area of 250 m$^2$ (50 m × 5 m); (2) smaller but also mobile, conspicuous fishes in an area of 150 m$^2$ (30 m × 5 m); (3) small, site-attached fishes within an area of 30 m$^2$ (30 m × 1 m); and, finally, (4) small cryptobenthic or larger fishes hidden in holes, also within an area of 30 m$^2$ (30 m × 1 m). All surveys were performed in slope or crest habitats, between 2 and 15 m depth, and by the same observer (RAM). In both Raja Ampat and Lizard Island, all surveys took place on fringing reefs around high islands, or on the exposed facet of lagoon-forming reefs, and thus encompassed exclusively upstream reef zones. Fish sizes were estimated with 2 cm precision for fishes under 30 cm TL and with 5 cm precision for all other fishes. All surveys were done between January and February and between October and December 2018. In Ha'apai, surveys were performed in the same types of reefs and reef habitats as Raja Ampat and Lizard Island and followed similar procedures, except that they encompassed only 2 phases. Phase 1 in Ha'apai targeted the same types of fishes as phases 1 and 2 in Raja Ampat and Lizard Island (in an area of 150 m$^2$, 30 m × 5 m), while phase 2 targeted the same types of fishes as phases 3 and 4 in Raja Ampat and Lizard Island (in an area of 60 m$^2$, 30 m × 2 m). All surveys in Ha'apai were performed by the same observer (PSW) between September 2017 and November 2018. A complete description of the procedures can be found in [75]. Overall, the regional dataset comprised 325 surveys in 3 locations.

We used a resampling procedure to scale counts from the different types of surveys and different count phases in each type of survey to the same common area (as in [11,43]). First, for each of the 4 (Raja Ampat and Lizard Island) or 2 (Ha'apai) phases of each survey, fish abundance was proportionally rescaled from the area surveyed (e.g., 250, 150, 60, or 30 m$^2$) to a common ("resampling") area of 100 m$^2$. Then, at each resampling iteration, a number of fish individuals equal to the rescaled abundance were randomly resampled for each phase of each survey. Thus, at each iteration, each final resampled survey (i.e., 100 m$^2$) consisted in the final, cumulative set of fish individuals resampled from all phases of the original survey. We refer to these resampled surveys throughout the manuscript as "survey areas," which comprise a smaller spatial scale compared to RLS sites. A total of 1,000 bootstrapping iterations were performed and all indicators with the exception of abundance (species richness, biomass, productivity, etc.) calculated as the mean across iterations. Total abundance was the criterion guiding the resampling principle, and, therefore, surveys had the same total abundance (but different species-specific abundance) across all iterations.

## Defining trophic groups and plankton subsidies

Species from both datasets were classified in trophic groups following the species lists provided in [11,44]. Five broad groups were used: herbivores (including macroalgivores, microphages, grazers, and detritivores, HD), omnivores (OM), invertivores (including mobile and sessile invertebrate predators, IN), zooplanktivores (PK), and generalised macrocarnivores (including specialised piscivores and large-bodied carnivores that often consumer fishes, cephalopods, and large crustaceans, GC [44]). Here, we consider plankton subsidies to be equivalent to pelagic subsidies, and we measure plankton subsidies as the net biomass production of planktivorous reef fishes. This definition of plankton/pelagic subsidies is narrower than the one used

previously (e.g., [11]) but was required because of the vast number of species with uncertain or unavailable quantitative dietary information, as well as the large spatial scale of our study. This narrow definition excludes indirect trophic links that may also result in pelagic subsidies being incorporated by reef communities (e.g., preying on planktivorous fishes or heterotrophic corals, coprophagy of planktivore waste, feeding on plankton-enriched detritus [40,50,54,55]). Conversely, this definition may not completely exclude reef-based material consumed by planktivores (e.g., resuspended algae material, fish, and crustacean eggs, emerging zooplankton [39,73,76]). While some of these links may be expected to decrease, others are expected to boost the perceived importance of pelagic material for reef fish assemblages.

There is strong evidence, however, that our estimates of plankton/pelagic subsidies are conservative. First, many of the "planktivorous" fishes that ingest substantial amounts of algae matter (e.g., *Abudefduf*, species of *Pomacentrus*, *Melichthys niger*, *Melichthys vidua*) were herein classified as "omnivores," rather than planktivores. These, therefore, do not contribute towards our results. Some of these fishes comprised a significant proportion of the productivity, as exemplified by 3 sites in Raja Ampat with 24.7%, 35.1%, and 43.3% of the productivity comprised exclusively by *Melichthys* spp. If included, the overall contribution of plankton to fish productivity in these cases would almost certainly be significantly higher. Second, other sources of reef-based matter, such as fish or crustacean eggs, often represent a minor to small proportion of the diet of diurnal planktivorous reef fishes (0% to 20% in most cases) [39,76–80]. Third, even nocturnal planktivores (e.g., holocentrids of the genus *Myripristis*, some apogonids), which feed to a higher degree on reef-based eggs and larvae and, particularly on resident zooplankton (primarily mysid shrimps), also feed extensively on pelagic copepods and other pelagic zooplankton [76]. Finally, some of the primary prey of nocturnal planktivores are thought to feed partially on pelagic-derived phytoplankton themselves (see p. 95 of [81]), which would make their predators secondarily reliant on pelagic sources, rather than reef sources. Therefore, it is likely that we provide an underestimate of pelagic subsidies—the real extent and magnitude of their energetic contribution to fish communities will probably be substantially higher than detected here.

### From standing biomass to fish productivity: Combining growth and mortality trajectories

We obtained length–weight conversion parameters (to estimate standing biomass) and growth and mortality coefficients (to estimate productivity) for all species in the regional dataset following the procedures in [82,83]. Briefly, species-specific Bayesian length–weight regression parameters were obtained from FishBase [84,85] and used to estimate individual weight (mass) for all fishes detected in the surveys. Then, mean sea surface temperature (calculated from remote sensing data, provided in the Bio-Oracle database [113]), species maximum body length ($L_{max}$), dietary and habitat use information were used to predict a standardised Von Bertalanffy Growth Model (VBGM) parameter ($K_{max}$) for each species at each location [83]. Both $L_{max}$ and $K_{max}$ were then used, alongside sea surface temperature, in Pauly's empirical regression to predict the instantaneous exponential natural mortality coefficient Z for each species, again, at each location [86]. This annual, population-level, mortality rate coefficient was then converted to a daily rate and combined with a risk function that describes the ontogenetic decline in mortality risk as individuals age and grow [82]. This resulted in an individual-level daily probability of mortality.

With growth trajectories and mortality probabilities defined for each individual fish at each location, productivity was then estimated under the individual age framework [82]. Expected somatic growth was quantified by (1) estimating the operational age of each individual fish at

the time of the survey [82,87]; (2) positioning the fish in their predicted VBGM trajectory according to their operational age and obtaining their size (mass) after a 1-day growth period; and (3) subtracting their mass at the survey from their forecasted mass after growth [82]. Finally, each individual was assigned a fate at the next day after the survey, mortality, or survival, simulated randomly according to their probability of mortality. Productivity was, finally, the expected growth of all surviving individuals at each bootstrapped simulation. Given the short time interval considered (i.e., 1 day), the terms "productivity" (i.e., rate) and "production" (i.e., the expression of this rate throughout a finite time interval) are used interchangeably throughout the text. The reasoning behind and the equations underlying each step described above can be obtained in detail from [82], which also provides the R software [88] package *rfishprod*, used for all the above calculations.

## Statistical modelling procedures

Except where otherwise stated, all modelling procedures in this study were implemented following the Bayesian framework and using the No-U-Turn Sampler (NUTS) with the Stan machinery, via the *rstan* interface to R and the *rstanarm* package [89,90]. Auxiliary priors (representing the shape parameter of gamma distributions and the phi parameter of beta distributions) were always defined following a Cauchy distribution with location = 0 and scale = 5. All other priors are defined on a case-by-case basis below. To propagate the variability and uncertainty in response variables arising from the resampling distributions described above, we ran 1 Bayesian model per resampling iteration in a total of 1,000 models per response variable/ modelling procedure (below, unless otherwise stated). We used the compounded posterior distributions of the modelled parameters from these 1,000 bootstrapped models to estimate coefficients and 95% HDP intervals. By propagating variability and uncertainty in the response variables, bootstrapping our models made our approach conservative when compared to an alternative strategy, i.e., using resampling medians to run a single model per response variable. For all Bayesian bootstrapped models, we used 3 MCMC chains per model and 1,500 NUTS steps per chain, with 50% burn-in.

Following the equivariance property of MCMC sampling, quantities derived from parameters estimated via MCMC are treated as posterior distributions of parameters themselves [91]. Thus, we derive our inferences from ratios between coefficients, i.e., between intercepts or slopes of locations. All comparisons and the acronyms used to represent these ratios in figures are specified below.

To ensure the convergence of 1,000 Bayesian models per modelling procedure, we implemented 2 automated checks. These checks evaluated, for each model, whether (1) the effective sample size of all parameters was equal to at least 100 times the number of chains (as recommended in [92], in our case, 300 steps); and (2) all Gelman–Rubin $R_{hat}$ statistic values were below 1.05. Only models that passed these checks by fulfilling these 2 criteria were retained in the compounded posterior distributions. Below we also list, for each model procedure, the number of models retained.

## The trophic structure of Indo-Pacific reef fish assemblages and the productivity–biomass relationships of 3 key locations

We started by exploring the ocean-scale (RLS) dataset, evaluating the potential correlation between the relative abundance of each of the 5 trophic groups of reef fishes (HD, OM, IN, PK, and GC; see above) and total fish abundance. To estimate the relationship between the relative abundance of trophic groups, we fit a local polynomial regression smoother (LOESS). We used a near-unitary α parameter (>0.9) to minimise sensitivity to outliers. Noticing that

planktivores were the only group positively associated with high fish abundance and that both high abundance and high planktivore proportions seemed to bear some level of spatial clustering at or near the "core" IAA ([93,94]), we then investigated the 3 selected locations in our regional scale dataset. These 3 locations spanned a steep gradient in biodiversity, from the core IAA (Raja Ampat, Indonesia), the centre of marine biodiversity in the world [42], through the periphery of the IAA (Lizard Island, Great Barrier Reef), to the central Pacific (Ha'apai, Tonga), over 6,000 km away from the core IAA.

We first examined abundance distributions for all species detected (i.e., not only planktivores) in the 3 locations. This unveiled striking distinctions in the relative status of abundant species between the locations. These patterns were then further explored by contrasting total fish abundance, $\alpha$, $\beta$, and $\gamma$ diversities, size and trophic structure, and productivity–biomass (PB) relationships between locations [43]. The median number of species per resampled survey (scaled as spp. 100 m$^{-2}$) was our metric of $\alpha$ diversity. We used the multivariate dispersion between resampled surveys (betadisper [95]) as our metric of $\beta$ diversity, with a nonmetric multidimensional scaling (nMDS) on a Bray–Curtis species dissimilarity matrix used to help visualise patterns. Finally, $\gamma$ diversity was estimated as the cumulative number of species in resampled surveys using sample-based rarefaction curves with the Coleman method [96,97].

Comparisons between localities (Raja Ampat, Lizard Island, and Ha'apai) were done via permutation tests (in betadisper), overlapping of 95% confidence intervals of estimates (rarefaction curves) or via Bayesian generalised linear models (GLMs). Bayesian GLMs had negative binomial (for total abundance and $\alpha$ diversity) or gamma (for standing biomass and productivity) error structures and used log link functions in both cases, with the following model structure:

$$y_s \sim NB(\lambda_s, \phi) \text{ or } y_s \sim Gamma(\alpha_s, \beta_s)$$

$$\ln\left(\frac{\lambda_s \phi}{1-\phi}\right) \text{ or } \ln\left(\frac{\alpha_s}{\beta_s}\right) = \beta_0 + \beta_1 x_{locality,s}$$

$$\beta_{0,1} \sim Normal(0, 100) \text{ or } \beta_{0,1} \sim Normal(0, 5)$$

where $y_s$ were the observed values of each indicator for each survey area $s$; $\frac{\lambda_s \phi}{1-\phi}$ and $\frac{\alpha_s}{\beta_s}$ represent the mean of the negative binomial and gamma distributions, respectively; $\beta_0$ is an overall intercept; and $\beta_1$ is an effect (intercept) for each locality, both of which with weakly informative, normally distributed priors. Differences between localities were summarised by the ratio of the posterior distributions of locality intercepts (i.e., $\beta_0 + \beta_1$) between Raja Ampat and Lizard Island (hereafter β$_{RL}$) or between Raja Ampat and Ha'apai (hereafter β$_{RH}$). As total fish abundance was the criterion used to inform the resampling procedure described above (see Study design and survey datasets), abundance was the same for each survey across all resampling iterations. Therefore, only 1 model (instead of 1,000 bootstrapped models) with 3 MCMC chains, 5,000 steps per chain, a thinning of 1 every 3 steps, and a 50% burn-in was used to compare total fish abundance between localities. For the other 3 comparisons among localities (i.e., species density, biomass, and productivity), all 1,000 models used parameters as above and passed the convergence checks, being thus retained for the compounded posterior distributions.

PB relationships describe variability in relationships between standing biomass and productivity and have been shown to be linked with reef fish turnover and exploitation rates [43]. We defined a model for PB relationship as:

$$y_s \sim Gamma(\alpha_s, \beta_s)$$

$$\ln\left(\frac{\alpha_s}{\beta_s}\right) = \beta_0 + \beta_1 x_{\mathrm{locality},s} \times \beta_2 x_{\mathrm{biom},s}$$

$$\beta_{0,1,2} \sim \mathrm{Normal}(0,5)$$

where $y_s$ represents the observed values of productivity in each survey area $s$; $\beta_0$ is an overall intercept; $\beta_1$ is an effect (intercept) for each locality; and $\beta_2$ is an effect (slope) for the standing biomass values of each survey area. We also compared PB relationships between localities using ratios. However, in this case, $\beta_{RL}$ and $\beta_{RH}$ are the ratios between the posterior distributions of the slopes of the curves in Raja Ampat and Lizard Island, and Raja Ampat and Ha'apai, respectively. For PB relationships, 994 out of 1,000 bootstrapped models passed the convergence checks and were retained for the compounded posterior distributions.

Size structure was investigated using biomass and productivity size spectra. These consisted of the cumulative biomass or productivity per fish size class (body mass in g) in $\log_{10}$ bins. We visually examined the overlap between size-spectra resampling quantiles for each size bin. Trophic structure was first also visually evaluated with an nMDS using a square root and Wisconsin-transformed, Bray–Curtis dissimilarity matrix calculated from the mean productivity of the 5 trophic groups at each resampled survey. This revealed a clear separation in the dots, particularly between those of Raja Ampat and the other 2 localities, which appeared to be driven by associations with distinct trophic groups. We then quantified these associations using Pearson's correlations. We calculated Pearson's $r$ values as the slopes of ordinary least squares linear regressions between the standardised productivity of each trophic group and standardised total productivity [98] at each locality, with intercepts fixed at 0:

$$\sum x_s = r_t x_{t,s}$$

where $\sum x_s$ is the total productivity of each survey area $s$; $x_{t,s}$ is the productivity of each trophic group $t$ at each $s$; and $r_t$ is Pearson's $r$ for each $t$. Prior to standardisation, all variables were $\log_{10}$ transformed to decrease dispersion among data points. All correlations were calculated for each resampling iteration assemblage, resulting in bootstrapped distributions of Pearson's $r$ that were analogous to the compounded posterior distributions of the bootstrapped Bayesian models above. We summarise these distributions with their 95% quantile intervals.

After finding a steep and tight relationship between planktivore productivity and total productivity (see main text), we also looked at the overall relationship between the proportional productivity of planktivores (predictor) and total productivity (response) using Bayesian generalised models. Because there was no reason to expect a specific shape for this relationship, we fit a series of models with different functions of proportional planktivore productivity: (1) a thin plate spline function (within a generalised additive model); (2) a log-linear function; (3) a second-degree polynomial function; and (4) a third-degree polynomial function. We compared these models using leave-one-out cross validation (using the package *loo* [99]), which showed that the best model (i.e., with the highest expected log predictive density (ELPD)) was model 3 ($\mathrm{ELPD}_{\mathrm{Model3}} = -475.9$; $\mathrm{ELPD}_{\mathrm{Model1}} = -477.6$; $\mathrm{ELPD}_{\mathrm{Model2}} = -532.7$; $\mathrm{ELPD}_{\mathrm{Model4}} = -476.8$). This model had the form:

$$y_s \sim \mathrm{Gamma}(\alpha_s, \beta_s)$$

$$\ln\left(\frac{\alpha_s}{\beta_s}\right) = \beta_0 + \beta_1 x_{\mathrm{pPK},s} + \beta_2 x_{\mathrm{pPK},s}^2$$

$$\beta_{0,1,2} \sim \text{Normal}(0, 10)$$

where $y_s$ represents the observed values of total productivity for each survey area $s$; $\beta_0$ is an overall intercept; and $\beta_1$ and $\beta_2$ are the linear and quadratic effects, respectively, of $x_{pPK}$, the proportional planktivore productivity of each $s$. Here, the emphasis was on the shape of the relationship between predictor and response, rather than on the exact values (or boundaries) of the coefficients, and, therefore, we fitted only 1 model (i.e., instead of bootstrapping 1,000 models) using the median values of total productivity and relative planktivore productivity across the 1,000 resampling iterations. We used 3 MCMC chains, 5,000 steps per chain, a thinning of 1 every 3 steps, and a 50% burn-in. We also calculated Bayesian $R^2$ values for each step following [100], which we summarise with their median and 95% HPD.

## Integrating datasets and predicting Indo-Pacific patterns of planktivore contribution to fish productivity

Finally, we integrated the ocean-scale, Indo-Pacific wide RLS dataset and our region scale, high-definition surveys from Raja Ampat, Lizard Island, and Ha'apai. To do that, we first modelled the effects of a series of potential predictors of the proportional contribution of planktivorous fishes to total fish productivity in the region-scale dataset. We then performed model selection (described below) to narrow down a final set of variables, which were used to predict the likely contribution of planktivores to the fish productivity of 1,031 sites that included planktivores, out of the total of 1,035 sites spread across most of the Indian and Pacific Oceans. Three sites from 1 ecoregion had unusually high values of one of the predictors and were subsequently excluded (see below). Thus, the final ocean-scale dataset used to predict planktivore contribution to total productivity consisted of 1,028 sites in 31 ecoregions.

Our initial set of potential predictors of planktivore relative productivity included 4 distinct classes: species-level predictors (average maximum species size, average species growth potential, and average species body shape), community-level predictors (planktivore abundance, species richness, and species composition), environmental predictors (bathymetric slope, mean and maximum surface current velocity, mean pelagic net primary productivity, and survey depth), and a proxy for human impacts (gravity of human impacts). These variables combine our knowledge of the drivers of fish biomass production (e.g., size, growth, overexploitation [43,101]), zooplankton availability flow (i.e., current speeds, phytoplankton productivity [14,39,73]), and physical constraints imposed by the energetic expenditure of swimming against the flow (e.g., [102,103]). Some of these predictors have also been found to drive the abundance and biomass of zooplanktivorous fishes across large spatial scales (e.g., [21,22]).

Species-level predictors were sourced from the same datasets used to estimate biomass and productivity (see From standing biomass to fish productivity: Combining growth and mortality trajectories above). Maximum species size and growth potential consisted of maximum length and $K_{max}$ parameter values for each individual. Species body shape was the standardised length–weight parameter $a$, which should increase in value as species change from slender and elongated to rotund, deep-bodied [83,104]. Average values of these variables were obtained by aggregating across all individuals for each survey. Community-level predictors consisted of the following: (1) the abundance of all planktivorous species in each survey; (2) number of species of planktivore species in each survey; and (3) taxonomic composition. Taxonomic composition was quantified by the 2 main axes (2 orthogonal variables) of a principal coordinate analysis (PCoA) on a Bray–Curtis dissimilarity matrix of family-level abundances at all sites.

Abundance data were fourth root and range transformed prior to the analysis. Following molecular phylogenetic evidence, we considered Caesionidae as part of Lutjanidae [105] and Microdesmidae as part of Gobiidae [106]. Analyses used the R package *vegan* [107].

An estimate of the "gravity of human impacts" for each site was obtained from the global dataset provided by [101]. We first converted the shapefile to a raster and then extracted values using the bilinear method, which interpolates the value of each point coordinate from its 4 nearest neighbours [108].

Finally, bathymetric slope, mean and maximum current velocity, and pelagic primary productivity were obtained from global rasters using the package *sdmpredictors* [109], which provides an interface between R and the Bio-Oracle and MARSPEC databases. Bathymetric slope is derived from the SRTM30_PLUS V6.0 [110,111], a 30 arc-second resolution, global digital elevation, and seafloor topography model). Surface current speeds are derived from the ORA-P5-ECMWF [112,113], a 1° resolution global ocean model driven by atmospheric forcing fluxes. Surface pelagic NPP is obtained from PISCES-SEXTANT-IFREMER [113,114], a global 0.25° resolution biogeochemical model. Mean and maximum current velocities and pelagic NPP are monthly averages for the period 2000 to 2014 and were interpolated and downscaled by Bio-Oracle [113], using a kriging model to achieve a final resolution of 5 arc-minutes. Bathymetric slope, mean, and maximum current velocity values for each site were extracted from a buffering area with a radius of 10 km around each site, while surface pelagic NPP was extracted from a buffering area with a radius of 30 km around each site. All extractions were accomplished using the *raster* package [108] in R.

Before model selection, variables observed to be overdispersed were $\log_{10}$ transformed, and Pearson's correlations were used to assess the potential for multicollinearity. This exposed high correlations between planktivore abundance, species richness, and the first PCoA axis ($r > 0.6$ for all combinations, $r = 0.80$ between abundance and richness); species average maximum size and $K_{max}$ ($r > -0.76$); and maximum current velocity and pelagic primary productivity ($r = 0.91$). Thus, we immediately excluded planktivore species richness, the first PCoA axis, species average $K_{max}$, and maximum current velocity. All remaining variable comparisons had $r < 0.5$. Model selection was performed on frequentist beta regression candidate model structures using the package *betareg* [115], with tools from *MuMIn* [116] in R. After narrowing down the model structure, the final model was refitted as a Bayesian beta regression following:

$$y_s \sim \mathrm{Beta}(\alpha_s, \beta_s)$$

$$\mathrm{logit}\left(\frac{\alpha_s}{\alpha_s + \beta_s}\right) = \beta_0 + \beta_1 \log_{10} x_{\mathrm{abun},s} \times \beta_2 \log_{10} x_{\mathrm{size},s} \times \beta_3 \log_{10} x_{\mathrm{curr},s} \times \beta_4 \log_{10} x_{\mathrm{NPP},s}$$

$$\beta_{0,1,2,3,4} \sim \mathrm{Normal}(0, 10)$$

where $y_s$ represents the proportional planktivore productivity in each survey area $s$; $\frac{\alpha_s}{\alpha_s + \beta_s}$ is the mean of the beta distribution; $\beta_0$ is an overall intercept; and $\beta_1$ to $\beta_4$ are effects (slopes) for $\log_{10}$-transformed planktivore abundance ($\beta_1$), average species maximum size ($\beta_2$), mean current speed ($\beta_3$), and pelagic primary productivity ($\beta_4$). For this predictive model of relative planktivore productivity, all 1,000 models passed the convergence checks and were retained in the compounded posterior distributions. We also calculated the Bayesian $R^2$ for this modelling procedure by aggregating $R^2$ values for all models in a compounded posterior distribution.

Despite the magnitude of differences in the geographic coverage and sample size between the ocean and regional datasets, heterogeneity in the range of the predictors and taxonomic composition were minor (**S1 Text, S7 Fig**), except for 3 sites that had unusually high pelagic

primary productivity (i.e., $>0.08$ g m$^{-3}$ day$^{-1}$, or more than 4 times over that of any other site). Given that these sites were located in a major urbanised area (Hong Kong), and hence likely subject to eutrophic conditions, these 3 sites were excluded before the predictive model above. Furthermore, a pairwise comparison between predicted proportional planktivore productivity in the ocean-scale dataset and empirically obtained values (region-scale dataset) revealed (1) substantial agreement for Raja Ampat and Lizard Island; and (2) similar mean values, despite a narrower range with lower maximum estimates in the ocean dataset for Ha'apai, likely to make our model conservative (**S1 Text**). Therefore, we used posterior linear prediction with this model's coefficients to predict the likely contribution of planktivores to the productivity of each site in the large-scale dataset. These site-level posterior predictions of planktivore relative productivity were summarised by their median values.

## Supporting information

**S1 Fig. Contrasting α, β, and γ diversity patterns do not explain extensive differences in fish abundance between reefs surveyed in Raja Ampat (Indonesia), Lizard Island (GBR), and Ha'apai (Tonga).** The top right inset shows a map with the sites sampled in each of the 3 locations. The species abundance relationship indicates only the 70 highest-ranking species at each location, with planktivores indicated by a black dot (for clarity, only planktivores among the 40 highest-ranking species displayed). Circles and intervals in the total fish abundance (top left inset) and α diversity (measured as sample-specific species density) represent Bayesian estimates and 95% HPD intervals, respectively. $\beta_{RL}$ is the ratio between model estimates in Raja Ampat and Lizard Island, and $\beta_{RH}$ between Raja Ampat and Ha'apai. Ribbons in the plot of γ diversity (measured from species rarefaction curves) are the 95% confidence intervals, while the dashed line represents the rarefaction point. The plot of β diversity (measured as the location-specific multivariate dispersion; see main text) shows the median and 95% quantile range of permutated multivariate dispersion. Map source: Natural Earth via the maps and mapdata packages in R. Numerical values underlying this figure are provided in "Morais_et_al_ FigS1.R," available from https://doi.org/10.5281/zenodo.5540102. GBR, Great Barrier Reef; HPD, high posterior density.
(TIF)

**S2 Fig. Distinct reef fish PB relationships among the 3 locations examined arise from similar size structures but different trophic structures. (A)** Regression lines for all locations had a similar slope but different intercepts. Dots are the bootstrap medians for survey areas, and segments delimit the 95% bootstrap quantile intervals (see Methods). Coloured lines represent 500 draws from the Bayesian posterior distributions, and black lines represent the posterior medians. **(B)** Biomass and **(C)** productivity size distributions, with median (dots) and 90% bootstrap quantile intervals. **(D)** An nMDS biplot showing trophic groups as vectors and survey areas as circles scaled proportionally to total fish productivity. **(E)** Relationship between the productivity of planktivores and total fish productivity in the 3 locations. Trendlines depict 300 bootstrapped Pearson's correlation r values (coloured lines) and median r (dashed lines) back calculated from standardised variables to productivity units (see Methods and S3 Fig) for each location. Circles are survey areas. Numerical values underlying this figure are provided in "Morais_et_al_Fig 03_FigS2.R," available from https://doi.org/10.5281/zenodo.5540102. GBR, Great Barrier Reef; GC, generalised carnivores; HD, herbivores/detritivores; IN, invertivores; nMDS, nonmetric multidimensional scaling; OM, omnivores; PB, productivity–biomass; PK, planktivores.
(TIF)

**S3 Fig. The relationship between the productivity of distinct trophic groups and total fish productivity for Raja Ampat, Indonesia; Lizard Island, on the GBR; and Ha'apai, in Tonga.** Raja Ampat: yellow, top row; Lizard Island: green, middle row; Ha'apai: blue, bottom row. Silhouettes denote, from left to right, planktivores, herbivores, omnivores, invertivores, and generalised carnivores. Trendlines depict 300 bootstrapped Pearson's correlation r values (coloured lines) and median r (dashed lines). Labels provide the median and 95% quantile interval of r values for each trophic group, at each location. Notice the standardised, log-scale in the axes of the panels. Numerical values underlying this figure are provided in "Morais_et_al_FigS3.R," available from https://doi.org/10.5281/zenodo.5540102. GBR, Great Barrier Reef.
(TIF)

**S4 Fig. The overall relationship between the proportional productivity of planktivores and total fish productivity across the 3 locations investigated.** Circles are survey sites. Thin black lines are predictions from 500 draws, and the red line is the median prediction from the posterior distribution of the model coefficients. $R^2$ values are the median and 95% high posterior distribution interval (in square brackets). Numerical values underlying this figure are provided in "Morais_et_al_FigS4.R," available from https://doi.org/10.5281/zenodo.5540102.
(TIF)

**S5 Fig. The relationship between the proportional productivity of planktivorous fishes and their abundance in Raja Ampat, Lizard Island, and Ha'apai.** In (A), circles are proportional to the x-axis and represent survey sites. (B) Proportional planktivore productivity as a function of mean species size, (C) site-specific mean surface current velocity, and (D) and pelagic NPP. Thin lines are 500 draws from the posterior distribution of the coefficients describing the relationship, and thick lines describe the median predictions. Notice the log-scale in the x axis of the panels. Numerical values underlying this figure are provided in "Morais_et_al_FigS5.R," available from https://doi.org/10.5281/zenodo.5540102. NPP, net primary productivity.
(TIF)

**S6 Fig. Gravity of human impacts among locations and the relationship between the productivity of planktivores and predatory fishes.** (A) The overall range of gravity of human impacts among sites in Raja Ampat and Ha'apai was similar but did not vary at Lizard Island, where it was minimal. (B) and (C) Modelling the potential effects of gravity on biomass (B) or productivity (C) showed a complete lack of relationship. Indeed, model selection procedures showed that a model including only locality performed better than any models including gravity. (D) The relationship between the productivity of generalised predators and the productivity of planktivores. This model tests for potential cascading effects from predator removal generating the trophic release of planktivores. Descending curves would provide evidence for this potential confounding process, yet the relationship was positive for the 3 localities. Thin lines are 400 draws from the posterior distribution of the coefficients describing the relationship, and thick lines describe the median predictions. Notice the log-scale in the x- and y-axis of the panels in (D). Numerical values underlying this figure are provided in "Morais_et_al_FigS6.R," available from https://doi.org/10.5281/zenodo.5540102.
(TIF)

**S7 Fig. The distribution of predictors of relative planktivorous reef fish productivity between the ocean and the region-scale dataset.** Coloured dots represent each site surveyed at each dataset, black circles represent mean values, and intervals represent the range. Notice that the y-axis is in the log-scale for planktivore abundance, and in the square root scale for mean current speed, species size, and mean pelagic NPP. These transformations were

introduced to increase resolution of the lower values. The taxonomic composition ordination represents the 2 main axes of a PCoA, with polygons as the maximum convex polygon of each dataset. Numerical values underlying this figure are provided in "Morais_et_al_FigS7.R," available from https://doi.org/10.5281/zenodo.5540102. NPP, net primary productivity; PCoA, principal coordinate analysis.
(TIF)

**S1 Table. Bayesian coefficients of the relationship between total reef fish standing biomass (predictor) and productivity (response) from Raja Ampat, Lizard Island and Ha'apai.** Estimates and variability are from the compounded chain including 1,000 Bayesian models run using Stan with the NUTS. Numerical values underlying this figure are provided in "Morais_et_al_Fig 03_FigS2.R," available from https://doi.org/10.5281/zenodo.5540102. HPD, high posterior density interval (95%); NUTS, No-U-Turn Sampler.
(DOCX)

**S2 Table. Bayesian standardised coefficients of the relationship between planktivorous fish abundance, maximum species size, mean surface current velocity, and mean pelagic NPP (predictors) and the proportional productivity of planktivorous fishes (response) from Raja Ampat, Lizard Island, and Ha'apai.** Estimates and variability are from the compounded chain including 1,000 Bayesian models run using Stan with the NUTS. Numerical values underlying this figure are provided in "Morais_et_al_Fig 04.R," available from https://doi.org/10.5281/zenodo.5540102. HPD, high posterior density interval (95%); NPP, net primary productivity; NUTS, No-U-Turn Sampler.
(DOCX)

**S1 Data. Final dataset constituting the ocean-scale dataset, filtered from the RLS and including 1,035 sites in 32 ecoregions distributed across the Western Indian, Central Indo-Pacific, and Central Pacific Oceans.** RLS, Reef Life Survey.
(CSV)

**S1 Text. A detailed exploration of differences in species diversity between the locations, the contrast between regional and local drivers of high planktivore abundance and productivity, spatial constraints of coral reef fish productivity, potential effects of human exploitation on productivity, and the expected contribution of planktivores to total productivity in the ocean-scale dataset.**
(DOCX)

## Acknowledgments

This is contribution #24 from the Research Hub for Coral Reef Ecosystem Functions. We thank Victor Huertas, Pauline Narvaez, Joshua Phua, Sam Shu Quin, Sabrina Inderbitzi, and Sophie Gordon for help throughout data collection, and the Reef Life Survey for the online available data.

## Author Contributions

**Conceptualization:** Renato A. Morais, Alexandre C. Siqueira, David R. Bellwood.

**Data curation:** Renato A. Morais, Patrick F. Smallhorn-West.

**Formal analysis:** Renato A. Morais.

**Funding acquisition:** Renato A. Morais, Patrick F. Smallhorn-West, David R. Bellwood.

**Investigation:** Renato A. Morais, Alexandre C. Siqueira, Patrick F. Smallhorn-West, David R. Bellwood.

**Methodology:** Renato A. Morais, Alexandre C. Siqueira, Patrick F. Smallhorn-West.

**Project administration:** Renato A. Morais, David R. Bellwood.

**Resources:** Renato A. Morais, Alexandre C. Siqueira, Patrick F. Smallhorn-West, David R. Bellwood.

**Software:** Renato A. Morais.

**Supervision:** David R. Bellwood.

**Validation:** Renato A. Morais, Patrick F. Smallhorn-West.

**Visualization:** Renato A. Morais.

**Writing – original draft:** Renato A. Morais.

**Writing – review & editing:** Renato A. Morais, Alexandre C. Siqueira, Patrick F. Smallhorn-West, David R. Bellwood.

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
