## [Editor Report · Decision Letter 0]

26 Mar 2021

Dear Dr Morais, 

Thank you for submitting your manuscript entitled "Spatial subsidies drive sweet spots of marine biomass production" for consideration as a Research Article by PLOS Biology.

Your manuscript has now been evaluated by the PLOS Biology editorial staff, as well as by an academic editor with relevant expertise, and I'm writing to let you know that we would like to send your submission out for external peer review.

Please re-submit your manuscript within two working days, i.e. by Mar 30 2021 11:59PM.

Kind regards,

Roli Roberts

Senior Editor

PLOS Biology

---

## [Decision Letter · Decision Letter 1]

18 May 2021

Dear Dr Morais,

Thank you very much for submitting your manuscript "Spatial subsidies drive sweet spots of marine biomass production" for consideration as a Research Article at PLOS Biology. Your manuscript has been evaluated by the PLOS Biology editors, an Academic Editor with relevant expertise, and by three independent reviewers.

You’ll see that all three reviewers are positive about the research question and the dataset, but reviewers #1 and #3 raise some substantial and non-overlapping concerns that will need to be addressed. Rev #1 has issues with your decision to bootstrap Bayesian analyses (and several other analytical decisions) – his requests will entail you re-doing many of your analyses, though he expects your central claim to survive. Rev #3 has concerns around the treatment of current and historical fishing, which s/he feels are neglected and need to be considered; the Academic Editor agreed with the reviewer on this issue, and thinks that this should probably be addressed with additional analyses, if possible. Rev #2 has more modest concerns, though these include a question about considering biomass instead of abundance.

In light of the reviews (below), we will not be able to accept the current version of the manuscript, but we would welcome re-submission of a much-revised version that takes into account the reviewers' comments. We cannot make any decision about publication until we have seen the revised manuscript and your response to the reviewers' comments. Your revised manuscript is also likely to be sent for further evaluation by the reviewers.

We expect to receive your revised manuscript within 3 months. 

**IMPORTANT - SUBMITTING YOUR REVISION**

*Re-submission Checklist*

*Published Peer Review*

*PLOS Data Policy*

*Blot and Gel Data Policy*

Sincerely,

Roli Roberts

Roland Roberts

Senior Editor

PLOS Biology

rroberts@plos.org

REVIEWERS' COMMENTS:

Reviewer #1:

[identifies himself as James Robinson]

 Morais et al. use a large underwater census dataset to show that pelagic (planktonic) subsidies are a major contributor to coral reef productivity, and these subsidies enable some reef systems to have exceptionally high levels of local biomass production. This study has the potential to considerably advance our understanding of biomass production on reefs by demonstrates a novel empirical link between biomass hotspots and pelagic energy inputs. The study question builds on theory predicting the importance of energetic subsidies to reef systems. The dataset is impressive, and well-suited to answer this question. The manuscript is well-written with thoughtful and impactful figures.

I believe the overall study question, data and context can be key advances, and should be suitable for publication in PLoS Biology. However, there are several fundamental problems in the Bayesian approaches used here. The method issues need addressed and the results may change. 

First, I don't understand the justification for bootstrapping Bayesian models, as this seems to introduce potential problems with model convergence that cannot be inspected. For example, you cannot assess the Rhat, effective samples and convergence of each 1000 models (1 per bootstrap sample), and so cannot be confident that the compounded posterior contains 100% valid samples. I see no reason why you wouldn't fit one Bayesian model to each response metric, as you are already calculating the mean response metric across iterations (L311). (it is also advisable to use the median of a bootstrap sample as this is robust to outliers, rather than the mean).

I would suggest re-running each analysis on the median bootstrapped ecological value, thoroughly inspecting model diagnostics, and using those model posteriors for inference. Or if there is serious justification for your approach (I cannot see it, and you do not cite these approaches), then adding material that summarizes the R-hat and convergence estimates for all 1000 models is necessary. But fitting one model per metric would be safer, just as robust (likely more), easier to implement and clearer to understand. 

Second, you use the ratio of posterior distributions to infer differences in site-level intercepts. Differences between Bayesian posteriors can be assessed by summation or differencing (Hobbs & Hooten, Bayesian Models, pg. 195), which would provide a confidence interval or Bayesian p-value for the difference between locations (in this case). For example, in Fig. 2, the ratio of location-intercept posteriors does not show that there are different intercepts. Rather, you would infer differences by comparing the posteriors of location-intercepts (do intervals overlap?), or by estimating the difference between posteriors of location-intercepts. I don't follow that ratios of distributions allows you to estimate certainty intervals either, though you use these statistics throughout the results (L103-108).

Third, in two analyses you fit a regression to the equation trophic abundance ~ total abundance, and total productivity ~ planktivore productivity. In this model structure, Y (e.g. total abundance/prod.) is directly dependent on X (e.g. planktivore abundance/prod.), violating model assumptions. As in your other analyses, why not fit relative planktivore abundance/prod. ~ total abundance/prod.? This affects Figs. 3, S3, S4.

I realise these recommendations are significant and apply to most of your statistical analysis. I must stress that I expect the overall result - planktivores contributing to most reef fish productivity, and predicting reef sweet spots - will be robust. But I see little justification for bootstrapping Bayesian models (have you an example of someone doing this?). Taking posterior distribution ratios seems mathematically incorrect. It is impossible to assess how many invalid posterior samples occur in these compounded posterior distributions. 

Minor comments

L44 - study [23] did not find any correlation between primary productivity and fish biomass, but hypothesized that high biomass = subsidy effect. There is an important distinction here between planktonic subsidies to planktivores, and pelagic ecosystem subsidies to reef piscivores (Doug McCauley's papers). This distinction should be clearer throughout - that you are testing for planktonic energy input to planktivorous fishes, though other pelagic subsidies are also known to promote biomass of upper trophic levels through different pathways (mobile consumers, foraging etc.).

Figure 3E and 4B - I find that scaling point size by productivity unhelpful as it hides data. Colour is sufficient here, and we would better understand the number and distribution of sweetspots if points are the same size. E.g. panel C shows many more sites than B, but is this just because the large points in B obscure many sites? Same applies to Fig 3E, as many points appear to cluster together.

L150 - this is a great result and clever way to link your analyses across scales

L187 - can you discuss the potential for planktivore abundances to be over/under estimated in hyperabundant locations? How might this influence your results?

L200 - can you discount the contribution of other pelagic subsidies to top predators here? It's not clear that planktivores are 'the' driving force, or one of several mechanisms by which pelagic subsidies raise fish biomass.

L319 - this definition is key and should be noted in the main text and abstract to emphasise that spatial subsidies are bottom-up and planktonic. You do not consider spatial subsidies to other trophic levels

L323 - previous sentence says 'narrow definition', here it is 'coarse'. What is it?

L324 - isn't predation on planktivorous fishes explicitly considered when you measure fish productivity, and this is why planktivore productivity corresponds with high overall biomass production?

L338 - how minor? Can you give an estimate of the range of non-pelagic diet items in planktivorous reef fishes?

L339 - presumably your surveys also do not census nocturnal planktivore abundances. Of course this still leads to an underestimate of planktivore productivity, but please note this limitation in the survey methods section too. If nocturnal planktivores are not in the abundance dataset, then it is also not relevant that these species have a higher degree of non-plankton food items…because your productivity estimates exclude all nocturnal species?

L380 - what is the point cloud? Clearer to just say you fit a LOESS smoother to data to estimate relationship between relative trophic abundance and total abundance.

L462 - what do the PC axes represent about taxonomic diversity in this context? 

L471 - what resolution are surface current pixels? Important because you are assuming some area is representative of plankton input to the reef site.

L467-L474 - much more detail needed here. What units are the productivity and surface current metrics? What are the data underlying these layers? Are they time-averaged over seasons, years or decades? 

L492 - you are making model predictions, not forecasts (ie a time-series prediction)

L501 - how many models were discarded? This is potentially problematic if certain metrics were difficult to fit, causing the bootstrapped posterior varies in size between each response metric.

L856-867 - This is useful oceanography context in the sup mat. Can you also consider how much intra-annual variability in your surface speed and primary productivity metrics? This is potentially important as UVC surveys are conducted at different times of year depending on location. I realise you cannot fully account for seasonal variability in these global UVC datasets, but some discussion of seasonality seems highly relevant for understanding planktonic spatial subsidies. More detail in these two remote sensing variables is also required in methods here, this will help others to build on your work.

Reviewer #2:

[identifies himself as Matthew McLean]

PLOS Biology review:

Spatial subsidies drive sweet spots of marine biomass production

My overall opinion of this manuscript is positive, and I have very few criticisms. The authors did an excellent job in constructing this manuscript, which is thorough, detailed, and well written. As a reviewer, it is not often that I come across a paper this clean on the first submission, so bravo to the authors. Ultimately, this paper should make an important scientific contribution well suited for PLOS Biology.

That said, I am not at all supportive of the terminology "sweet spots." I see what the authors are going for here - nice try - but this is going too far, trying to push a new flashy term with no actual meaning or value. Let's look at the actual definitions of both "sweet" and "sweet spot" and see if they capture the main ideas of this paper. From Google definitions:

Sweet. Adjective:

1) having the pleasant taste characteristic of sugar or honey; not salty, sour, or bitter. 

2) pleasing in general; delightful

3) (of a person or action) pleasant and kind or thoughtful

4) used for emphasis in various phrases and exclamations

I am going to argue that none of these definitions align with the main ideas of this paper.

Sweet spot. Noun:

1) the point or area on a bat, club, or racket at which it makes most effective contact with the ball

2) an optimum point or combination of factors or qualities

Number 1 is a definite no. Number 2 is leaning toward what the authors are referring to in their paper, but high planktivore biomass is not necessarily "an optimum point."

So, I am going to argue that none of the available definitions of sweet nor sweet spot align with the main ideas of this paper. Moreover, let's look at antonyms of sweet. These include harsh, sour, bitter, disagreeable, nasty. So, if areas of high planktivore biomass are "sweet" then are areas of low planktivore biomass "sour" or "bitter"? And, by that nature, is Figure 4 showing us the sweet-to-sour planktivore index?

In all seriousness, simply replacing sweet spot by hot spot won't change anything about the paper or its quality and will be easily recognized and understood by any reader, so I highly encourage the authors to make this change.

I would also encourage the authors to consider changing "spatial subsidies" to "planktivore subsidies" or "plankton subsidies" in the title because "spatial subsidies" is not very informative. The "spatial subsidy" referred to here is an influx of plankton into the system that is captured by planktivorous fishes, leading to exceptional biomass production. Saying "planktivore subsidies" in the title would make this immediately clear for readers.

My other main comment/question is why was abundance used to assess trophic structure instead of biomass, especially if biomass production is the main focus of the paper? For instance, Figure 2 shows trophic structure based on abundance. Planktivores, being small-bodied, schooling fishes, might show exceptionally high abundances, but not necessarily exceptionally high biomasses. Certainly, biomass is a better descriptor of trophic processes than pure abundance, given that trophic dynamics refers to the transfer of energy and material through the food web? It seems that these results could change substantially with biomass instead of abundance, with a much lower contribution from planktivores. The authors should explain and justify why abundance was used here instead of biomass, and I apologize if I am missing something obvious.

Line 18: what makes these locations "key"?

Line 19: Figure 4 does not appear to show 2/3 of the global ocean. Do you mean 2/3 of the tropical ocean?

Line 28: flows of energy and transport of matter in and of themselves are not spatial subsidies, so this first sentence is written incorrectly. The proper definition, however, is given in the following sentence. You should rewrite or combine these first two sentences to give a very clear and explicit definition of what a spatial subsidy is.

Figure 1: great pictures! I'm jealous of Yen Yi Lee…

General comment: be consistent with the name Ha'apai, sometimes there is an apostrophe, sometimes not

Line 71: you chose 50% as the threshold to define hot spots, but why was this number chosen? This is glossed over very quickly, but its crucial to the entire paper. I can't seem to find the justification for this number anywhere in the manuscript.

Line 79: you should specify whether this is proportion of species or abundances/individuals.

Figure 2 panel B: the clear shift from invertivore-dominated to planktivore-dominated suggests a shift from benthic to pelagic energy channels, which is supported by the influence of slower currents and higher NPP. This might be worth mentioning in the discussion, particularly if a higher contribution of pelagic energy channels is key to supporting biomass hotspots. 

Line 123: HPD is for what interval/percentile?

Line 126: "occupied a largely different area of the productivity spectrum" is really vague, sounds overly technical, and is not very meaningful. You should rewrite this in a more precise manner to say exactly what the difference is (i.e., raja ampat is higher in both total productivity and planktivore productivity).

Line 146: I am not really sure what you mean here by "reconciling the energetic implications" try to use less ambiguous and more precise wording to help the reader follow and understand the flow of the storyline.

Line 151: maybe I am not understanding what you are saying here, but if all other trophic groups remain static and planktivore abundance goes up, isn't it obligatory that planktivores would account for more of the productivity, i.e., relative planktivore productivity would increase? If I am missing the idea of the sentence, please try to clarify.

Line 154: "would result in more than 50% of total productivity based on planktivores" I think you can rewrite this more clearly, something like "more than 50% of total productivity would be accounted for by planktivores"

Line 159: the phrase "planktivore subsidies" here immediately makes me think that the title would be more accurate and easier to interpret if it said "planktivore subsidies" or "plankton subsidies" rather than just "spatial subsidies" which by itself is not very informative.

Line 165: what's the definition of "pelagic subsidy" here? Is this identical to "planktivore subsidy" or is it distinct? If they are interchangeable, it might be better to stick with one term only for consistency.

Line 177: if these predictions were made under a Bayesian framework, why do you present means and standard deviations here instead of medians and credible intervals/HDP like elsewhere in the paper?

Line 179: is the 20% expected value a model estimate, or is this just saying that 100% divided equally by 5 trophic groups would yield 20% each? If the latter, perhaps this should be rethought because in reality, one would never expect equal representation across trophic groups.

Line 186: perhaps you should say "in almost any region" instead of "almost anywhere" to make the distinction between regions and sites. For instance, you show that these hot spots can arise in any region, but they aren't necessarily possible on any site within a region since they arise under certain environmental conditions. 

Line 189: somewhere in the manuscript it might be good to really stress the idea that as you add more and more fish to a system, it's basically just more and more planktivores. It might also be worth thinking about/discussing whether other trophic groups hit a ceiling, whereas the biomass ceiling for planktivores is much less limited.

Line 196: you haven't really touched on external productivity yet in the manuscript, it would be good to make this clear for the reader that you are referring to an influx of plankton, particularly in areas with slow currents and high NPP.

Line 215: you haven't spoken about nutrition yet in the paper, so what do you mean specifically here? Yes, greater fish biomass leads to greater energy flow/production, but if, for instance, planktivores are low in nutrient concentration, then there isn't really an expansion of the "nutritional footprint." Moreover, I would encourage you to use wording that is a bit more meaningful and less ambiguous than "energetic and nutritional footprint"

Line 380: "the point cloud" hasn't been defined yet, so you might want to give a more specific description of what you are referring to here (i.e., data were first plotted and then fitted)

Line 390: what were these striking distinctions?

Paragraph beginning on line 398: I would encourage you to provide the equations for the Bayesian models you performed

Moreover, because model equations are not provided, it is not clear to me at what scale each of the models were performed? Were the replicates transects, sites, or regions? Were the models nested with random effects? This needs to be specified. Additionally, it should be specified at which spatial scale the predictor variables are measured and enter these models. 

Line 422: here, again I am not sure what "trophic imprint" refers to, so you would be better off clearly stating the result you saw.

Line 441: here, I initially saw "after using model selection…" and thought that not nearly enough detail was provided on how model selection was performed. I then found that the subsequent paragraphs explain this procedure. I suggest rewriting this to make this very clear. For instance, something like, "We then performed model selection (described below) to narrow down a final set of variables, which were then used to predict the likely contribution of planktivores…"

Line 499: do you mean number of effective samples?

Line 812: the second and third "comprising" should be changed to "comprised"

Line 879: this should be "Assuming that 40-70% of this is consumed, ~50% of this being by herbivorous fishes…" Also, what is the basis for assuming 40-70% consumption?

Line 880: to avoid confusion with arithmetic mean, change this to "this would result in a maximum expected…"

Line 894: I encourage you to use a much simpler word than "impinges"

Line 909: what is the "energetic matrix"? Try to be very clear about what you mean here

Line 915: you need to state that you compared these things between the two datasets, otherwise it reads as if you just compared distributions of the four variables to each other

Figure S1: Haapai is cut off the map

I am signing my review for transparency and invite the authors to contact me if they have any questions.

Sincerely,

Matthew McLean

Reviewer #3:

This paper explores whether energetic subsidies from plankton-rich waters around some coral reefs are the basis for high fish abundance and productivity in those areas - and whether consequently, planktivores as a trophic group are the key drivers of high productivity on coral reefs. This is a really interesting paper with a conclusion that could be vital to coral reef management. The authors make excellent use of a valuable global dataset and the manuscript is well-written with really excellent figures. 

My main concern with the paper surrounds the potential impact of current and historical fishing on the data, and on the conclusions drawn. At no point in the manuscript is the effect of fishing on the observed abundance, productivity, size structure or trophic structure of the communities discussed, and yet fishing is likely to alter all of these metrics. Furthermore, the fishing pressure across the sites is likely to vary significantly at all of the spatial scales considered. To my mind, without accounting for fishing effects, it is very hard to draw conclusions about differences in trophic structure and dominant trophic groups between sites. For example, under high fishing pressure you would expect a decline in large carnivores, falling into the GC category - but with their removal, you might also expect an increase in both the abundance and the relative contribution to the community of lower trophic levels (their prey) - including small planktivores that contribute to the conclusions drawn in this study. It's also interesting to note from Figure 2 that arguably the most remote sites in terms of fishing pressure, in the middle of the Pacific, also seem to show a trend for reduced dominance of planktivores. Indeed, they appear to have a greater proportion of omnivores, herbivores and carnivores in general - which you might interpret as a more in-tact coral reef trophic structure. I think this is also particularly important because the sites with the greatest overall abundance in the coral triangle, are data points that make a disproportionally important contribution to the overall conclusion of this study - the very high productivity sites for example, are found here. This area is also arguably much more heavily fished, in general than locations in Australia or the central Pacific. Perhaps the authors have thought about this and selected data from sites with minimal / no fishing, in which case this justification needs to be clear. Otherwise, some metric of fishing pressure ought to be included in the analyses to quantify its contribution to the observed patterns in the data.

Also on fishing, I think it is important to include some information on the target species in these fisheries. The key message of the paper is that energy subsidies support "sweet spots" for reef fisheries but without qualifying whether the planktivores that make up this high productivity are targeted, or indeed available to fishermen on these reefs, the message is a little strong. I can fully believe that reefs in locations with high water flow or upwelling support abundant planktivore communities - but unless these fish are targeted, or contribute significantly to energy flow up the food chain to groups that are targeted, then I wonder if they are not really a separate ecosystem - at least from the perspective of the fishermen. Perhaps more information could be pulled into the main paper about the identities of the species contributing to the overall patterns - and the extent to which these are targeted - or consumed by predators that are.

I really like this paper, and do not fundamentally disagree with the conclusion, but I think this is a gap that really needs addressing before publication. I would also like to acknowledge that I am not well-versed in the Bayesian analyses employed here - and whilst to my understanding these analyses seem thorough and appropriate, it will be important to have the review of someone with more expertise to cover that part of the paper. Below I include some more minor comments to assist with re-working of the paper.

Introduction

Line 28: the first line suggests that flows of energy and transport of matter are a definition of spatial subsidies - I think this is misleading.

Figure 1 - the reference to this figure, and the legend that goes with it, I do not think is appropriate - they are nice images that convey the idea of concentrated fishing effort - and planktivore-dominated reef fish assemblages - but they don't provide evidence that fishers do not fish randomly - and panel B definitely doesn't show how plankton underpins sweet spots of fish productivity - it just shows aggregations of planktivores on a reef.

Line 64: dissecting is a strange choice of word - examining / disentangling perhaps?

Results

Lines 105 - 108: I find the reporting of the data confusing here - e.g. 60% and 206% higher in Raja Ampat compared with Lizard and Ha'apai - it's an effort to work out which numbers relate to which comparison - it would be clearer to be explicit about each.

Line 125 - I don't think that Figure 3D shows how carnivores comprise a fraction of the productivity of planktivores - perhaps this is supposed to point to a different panel?

Discussion

Line 187 - how are you defining over-representation? If it's over-represented you must be comparing it to some expectation of what it's representation should be?

Line 215 - Another thing that struck me was the assumption that the reef-planktivores captured by your surveys were obtaining their food from large areas - depending on the identity of these fish species, many are site-attached to reefs - relying on them for structure for example - and so not moving significant distances. Any those that are moving are arguably not part of the reef, or the reef fishery. Perhaps you only mean that their food comes from a larger area, and is transported to them, in their location on the reef - but this needs to be clear.

Supplementary material

Line 949-951 - I think it's really interesting here that you are finding "sweet spots" in areas where you don't expect them to occur due to high planktonic productivity - 20% is not a small amount of variation and so digging into what is happening in these systems would be really interesting. Detrital recycling is of course the other process by which high productivity is achieved on reefs with limited primary production. Would the dataset allow you to explore the role of that in your observed patterns?

Figure S4 - I may be misunderstanding the analyses here but I don't understand the fit drawn through the data in the Haapai panel - why is the line so shallow and the points much steeper?

---

## [Decision Letter · Decision Letter 2]

16 Sep 2021

Dear Dr Morais,

Thank you for submitting your revised Research Article entitled "Spatial subsidies drive sweet spots of marine biomass production" for publication in PLOS Biology. I have now obtained advice from the original reviewers and have discussed their comments with the Academic Editor. 

Based on the reviews, we will probably accept this manuscript for publication, provided you satisfactorily address the remaining points raised by the reviewers. Please also make sure to address the following data and other policy-related requests.

IMPORTANT:

a) Please attend to the remaining requests from reviewer #1.

b) We wonder if it might be more accurate to make the title slightly more specific, by including the word "tropical" ("sweet spots of tropical biomass") - entirely up to you, but we thought we'd raise this point.

c) Please address my Data Policy requests below; specifically, please supply numerical values underlying Figs 2AB, 3ABCDE, 4ABC, S1, S2ABCDE, S3, S4, S5ABCD, S6ABCD, S7, and cite the location of the data clearly in each relevant Fig legend. We'd strongly discourage the use of an institutional repository, on the grounds of long-term robustness; we had a recent case where a major US University changed their data URLs one month after we published a paper; this will now necessitate a published Correction.

We expect to receive your revised manuscript within two weeks. 

*Published Peer Review History*

*Early Version*

Sincerely,

Roli Roberts

Senior Editor,

rroberts@plos.org,

PLOS Biology

DATA POLICY:

Regardless of the method selected, please ensure that you provide the individual numerical values that underlie the summary data displayed in the following figure panels as they are essential for readers to assess your analysis and to reproduce it: Figs 2AB, 3ABCDE, 4ABC, S1, S2ABCDE, S3, S4, S5ABCD, S6ABCD, S7 NOTE: the numerical data provided should include all replicates AND the way in which the plotted mean and errors were derived (it should not present only the mean/average values).

DATA NOT SHOWN?

REVIEWERS' COMMENTS:

Reviewer #1:

[identifies himself as James Robinson]

The authors have been exceptionally thorough in dealing with my statistical concerns, and I appreciate the detailed responses and extra analyses now in the manuscript. The revised methods demonstrate the reasoning and robustness of the bootstrapped Bayesian approach (useful to see the comparison of HPDs in Table 1, thanks). The trophic productivity models are also now improved, and I think the new analyses of human gravity and piscivore productivity add great context.

Just a last thought regarding the overall implication that most reef fisheries benefit from planktonic sweet spots (L279-286), and the comment from R3 on target species. As you write, high biomass production will support specific fisheries - gears with selectivity for planktivores (nets) or consumers in higher trophic levels (handlines). But nets and traps also target lower trophic level species, like herbivores, which likely correlate with benthic productivity but not plankton sweetspots. Perhaps 'herbivore sweetspots' would be predicted by a different set of variables (algal cover, depth, coral cover, rugosity, L1137) that exist in different sites to plankton sweet spots (though are constrained by algal productivity). Regions with both sweetspots would be complementary - and perhaps best sustain diverse reef fisheries by offering resilience against ecosystem change (a portfolio effect). Even though herbivore productivity is far smaller than planktivore (L1134), herbivores dominate catch on many heavily fished reefs (Kenya, see Hicks & McClanahan 2011; Seychelles, my work with Nick Graham; Indonesia, Humphries et al. 2019). 

This idea is beyond the scope of your study. But you could note in the discussion that some key reef fisheries (herbivores) are unlikely to benefit/correlate with plankton sweetspots. I think this is important for considering how trophic group productivity will influence fisheries on degraded reefs (L290), e.g. in Morais et al. 2020 (Func. Ecol.) you showed that herbivores drive biomass and productivity increases on turf-dominated reefs.

The final figures are great, as usual from this team. One minor suggestion for Figure 3B, 3C + S2B,C - adding jitter/dodge to points would reveal error bars.

The study is a big advance for reef energetics + fisheries - I recommend acceptance and look forward to reading in print. 

Reviewer #2:

[identifies himself as Matthew McLean]

Great job on the revision, I look forward to seeing the paper in print.

Reviewer #3:

I have now reviewed the authors' responses to all of the reviewers' comments, in particular my own. I believe that the authors have done a really thorough job of dealing with the comments and that their manuscript is now greatly improved. I think this is ready for publication now and I think it will make an excellent contribution to the journal.

---

## [Editor Report · Decision Letter 3]

4 Oct 2021

Dear Dr Morais,

On behalf of my colleagues and the Academic Editor, Pedro Jordano, I'm pleased to say that we can in principle offer to publish your Research Article "Spatial subsidies drive sweet spots of tropical marine biomass production" in PLOS Biology, provided you address any remaining formatting and reporting issues. These will be detailed in an email that will follow this letter and that you will usually receive within 2-3 business days, during which time no action is required from you. Please note that we will not be able to formally accept your manuscript and schedule it for publication until you have made the required changes.

PRESS: We frequently collaborate with press offices. If your institution or institutions have a press office, please notify them about your upcoming paper at this point, to enable them to help maximise its impact. If the press office is planning to promote your findings, we would be grateful if they could coordinate with biologypress@plos.org. If you have not yet opted out of the early version process, we ask that you notify us immediately of any press plans so that we may do so on your behalf.

Sincerely, 

Roli Roberts

Roland G Roberts, PhD 

Senior Editor 

PLOS Biology

rroberts@plos.org